# GENERAL SEARCH TECHNIQUES WITHOUT COMMON KNOWLEDGE FOR IMPERFECT-INFORMATION GAMES, AND APPLICATION TO SUPERHUMAN FOG OF WAR CHESS

**Brian Hu Zhang**
MIT CSAIL
zhangbh@csail.mit.edu

**Tuomas Sandholm**
Carnegie Mellon University
sandholm@cs.cmu.edu

## ABSTRACT

Since the advent of AI, games have served as progress benchmarks. Meanwhile, imperfect-information variants of chess have existed for over a century, present extreme challenges, and have been the focus of decades of AI research. Beyond calculation needed in regular chess, they require reasoning about information gathering, the opponent's knowledge, signaling, *etc*. The most popular variant, *Fog of War (FoW) chess* (a.k.a. *dark chess*), has been a major challenge problem in imperfect-information game solving since superhuman performance was reached in no-limit Texas hold'em poker. We present *Obscuro*, the first superhuman AI for FoW chess. It introduces advances to search in imperfect-information games, enabling strong, scalable reasoning. Experiments against the prior state-of-the-art AI and human players—including the world's best—show that *Obscuro* is significantly stronger. FoW chess is the largest (by amount of imperfect information) turn-based zero-sum game in which superhuman performance has been achieved and the largest zero-sum game in which imperfect-information search has been successfully applied.

## 1 INTRODUCTION

The concept of breaking a large problem into subproblems and *searching* through them individually has been with us since time immemorial. In artificial intelligence (AI), search is a core capability that is required for strong performance in many applications. In game solving, this commonly takes the form of *subgame solving*. In games of perfect information, subgame solving is conceptually straightforward, because every new state induces a subgame that can be analyzed independently of the rest of the game. Subgame solving in perfect-information games is as old as computers themselves: Alan Turing and David Champernowne wrote a chess engine *Turochamp* in 1948 using minimax search and a hand-crafted function for evaluating nodes (Kasparov & Friedel, 2018). In landmark results, subgame solving has played a key role in reaching superhuman level in chess (Campbell et al., 2002) and go (Silver et al., 2016; 2017; 2018).

In contrast to such perfect-information games, most real-world settings are *imperfect-information* games. These include negotiation, business, finance, and defense applications. Thus it is crucial for the field of AI to develop strong techniques for imperfect-information games. Such games involve additional challenges not present in perfect-information games. For example, AIs for imperfect-information games might need to randomize their actions to prevent the opponent from learning too much information, and a player's optimal action in a state can depend on that same player's action in a totally different state. Therefore, subgame solving in imperfect-information games is drastically more difficult. Methods for real-time subgame solving in imperfect-information games have only been developed relatively recently (Gilpin & Sandholm, 2006; 2007; Waugh et al., 2009; Ganzfried & Sandholm, 2015; Burch et al., 2014; Moravcik et al., 2016; Brown & Sandholm, 2017; Moravčík et al., 2017; Brown & Sandholm, 2018; 2019; Brown et al., 2020; Sokota et al., 2024), and they were key to achieving superhuman performance in no-limit Texas hold'em poker (Brown & Sandholm, 2018; 2019; Moravčík et al., 2017). Strong AI performance has also been achieved

in a few imperfect-information zero-sum games that are even larger (Vinyals et al., 2019; Berner et al., 2019; Perolat et al., 2022). However, these were accomplished with learning alone and did not enjoy the further performance benefits that search could bring, due largely to the lack of scalability of subgame solving algorithms for imperfect-information games larger than poker.

In this paper we present dramatically more scalable general-purpose subgame solving techniques for imperfect-information games. We used these techniques to create *Obscuro*, an AI that achieved superhuman performance in *Fog of War (FoW) chess* (a.k.a. *dark chess*), the most popular variant of imperfect-information chess. Over 120 games against humans of varying skill levels—including the #1-ranked human—and 1,000 games against the previous state-of-the-art FoW chess AI (Zhang & Sandholm, 2021), we conclusively demonstrate that *Obscuro* is stronger than any other current agent—human or artificial—for FoW chess. FoW chess is now the largest (measured by amount of imperfect information) turn-based game in which superhuman performance has been achieved and the largest game in which imperfect-information search techniques have been successfully applied.

In the next section we will introduce the game and discuss the challenges that players in these types of games must tackle. In the section after that, we present our AI agent *Obscuro* and the algorithms therein. In the section after that, we present our experiments. Finally we present conclusions and future research directions.

## 2 CHALLENGES IN IMPERFECT-INFORMATION GAMES SUCH AS FOG OF WAR CHESS

Imperfect-information versions of chess have captured the imagination of chess players and scientists alike for over a century. To our knowledge, the first imperfect-information version of chess was *Kriegspiel*, invented in 1899 and based on the earlier game *Kriegsspiel*, a war game used by the Prussian army in the early 19th century for training (Pritchard, 1994). In the modern day, there are multiple imperfect-information variants of chess, including *Kriegspiel, reconnaissance blind chess* (RBC), and *Fog of War (FoW) chess*.[1] Imperfect-information chess is a recognized challenge problem in AI. Although there has been AI research in Kriegspiel (Parker et al., 2005; Russell & Wolfe, 2005; Ciancarini & Favini, 2009) and RBC (Gardner et al., 2023), strong performance has not been achieved in Kriegspiel, and RBC is not played competitively by humans. By comparison, FoW chess has surged in popularity due to its implementation on the major chess website chess.com, and strong human experts have emerged among thousands of active players.[2] It is the most popular variant of imperfect-information chess by far, and strong human experts exist who can serve as challenging benchmarks of progress.

FoW chess presents a unique combination of challenges that did not exist in prior superhuman AI milestones.[3] First, chess itself is a highly tactical game often requiring careful lookahead, and FoW chess is no different: there are often positions where one player has perfect or near-perfect information and can execute a sequence of moves that results in an advantage. Thus, a strong agent must have solid lookahead capability. Lookahead in other games is usually accomplished by subgame solving. Thus it would be desirable to be able to conduct subgame solving in FoW chess too.

Second, private information is rapidly gained and lost. It is possible for the size of a player's *information set* (infoset)—*i.e.*, set of indistinguishable positions given a player's observations—to rapidly increase and then decrease again, for example, from hundreds up to millions and then back down to hundreds, in a matter of a few moves. Thus, a strong agent must have the ability to reason about this rapidly-changing information.

Third, a strong agent must at least somewhat play a *mixed strategy*—that is, it must randomize its actions. Otherwise, an adversary who knows the strategy, or has learned the strategy from past observation, can easily exploit that knowledge.

---

[1]Despite its similar name, *Chinese dark chess* has no private information, and thus does not require the types of reasoning that are required in FoW chess.

[2]As of April 2025, the Fog of War chess leaderboard on chess.com (Chess.com, 2025a) listed 19,150 active players.

[3]The complete rules of FoW chess can be found in Appendix A

Finally, in games like FoW chess, reasoning about *common knowledge* is difficult. This is a key challenge because most algorithms for subgame solving—including those that led to breakthroughs in no-limit Texas hold'em poker—rely on the ability to reason about common knowledge, or often even the ability to *enumerate* the entire common-knowledge set—that is, the smallest set of histories $C$ with the property that it is common knowledge that the true history lies in $C$ (Brown & Sandholm, 2018; 2019). So, to prepare for solving a subgame, prior algorithms need to reason about what the agent knows about what the opponent knows about what the agent knows, and so on. This need can dramatically expand the set of states that need to be incorporated into the subgame solving algorithm, making such methods impractical for games much larger than no-limit Texas hold'em.

For example, consider the two FoW chess positions in Fig. 1.[4] Although seemingly completely distinct, it is possible to show (see Appendix E.1) that these two positions are connected by no fewer than nine levels of "I think that you think that..." reasoning. Prior techniques would require the ability to generate this complex connection before starting subgame solving from either of the two positions.

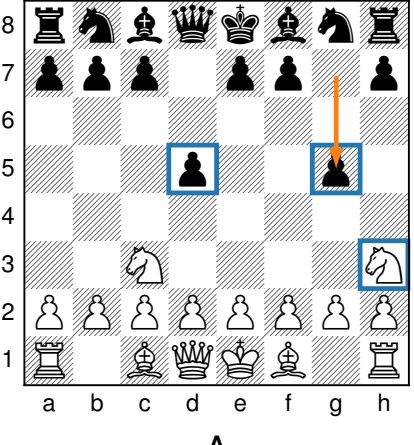 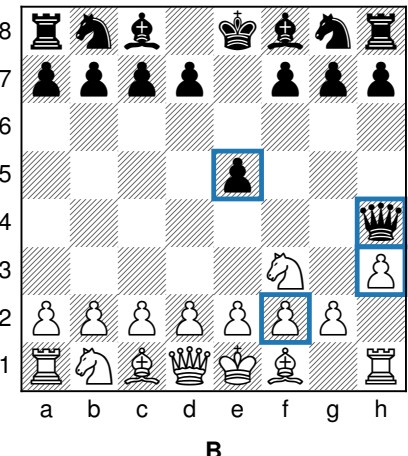

**A**  **B**

Figure 1: Two FoW chess positions in the same common-knowledge set. **(A)** position after moves **1. Nc3 g5 2. Nh3 d5**; **(B)** position after moves **1. Nf3 e5 2. h3 Qh4**. The boxed squares mark pieces visible to the opponent.

Such intricacies make it difficult to reason about common knowledge efficiently. For example, common-knowledge sets in FoW chess can quickly grow prohibitively large, so they cannot be held directly in memory (Zhang & Sandholm, 2021). In FoW chess, individual *infosets* often have size as large as $10^6$ and can have size $10^9$. Common-knowledge sets can have size $10^{18}$—far too large to be enumerated in reasonable time or space during search. (Detailed calculations for these lower bounds can be found in Appendix E.1.) Perhaps even more troubling is the fact that it is not even clear that it is possible to efficiently decide whether two histories can be distinguished by common knowledge, so in some sense reasoning about common knowledge may *require* enumerating the common-knowledge set in the worst case.[5]

This is in sharp contrast to poker, which has special structure that has driven the success of past efforts in that game. First, at least in two-player (heads-up) Texas hold'em poker, common-knowledge sets are not very large. They have size at most $\binom{52}{2}\binom{50}{2} \approx 1.6 \times 10^6$, and can thus easily be held in memory. Moreover, thanks to poker-specific optimizations (Johanson et al., 2011), subgame solving in poker can be implemented in such a way that its complexity depends not on the size of the common-knowledge set but merely on the size of the infoset, enabling feasible subgame solving even

---

[4]The sequence of moves in the figure is purely for the purpose of illustrating common knowledge, and does not represent strong play. For example, *Obscuro* never plays **1... g5** or **2... Qh4**.

[5]Solinas et al. (2023) formally state and study this and similar computational problems in general games, showing that they are intractable in the worst case, so it should perhaps not come as a surprise that they appear to be hard in FoW chess.

when the common-knowledge sets are large, as is the case in multi-player poker.[6] In more general games where these domain-specific techniques do not apply—such as FoW chess—the complexity of traditional subgame-solving techniques for imperfect-information games would scale with the size of the common-knowledge set, which in our case renders such techniques totally infeasible.

# 3 DESCRIPTION OF OUR AI AGENT *Obscuro* AND THE NEW ALGORITHMS THEREIN

The technical innovations of *Obscuro* are in its search algorithms. At a high level, they operate as follows. At all times, the program maintains the full set $P$ of possible positions[7] given the observations that it has seen so far in the game, as well as a partial game tree $\hat{\Gamma}$ consisting of its calculations from the previous move. At the beginning of the game, $P$ contains only the starting position $s_0$, and $\hat{\Gamma}$ consists of a single node $s_0$, since the program has done no calculation. Although $P$ is small enough to fit in memory (usually $|P| \leq 10^6$), it is too large to feasibly allow nontrivial reasoning about every single position in $P$ on every move. Therefore, the program instead samples a small subset $I \subseteq P$ at random, whose size is no more than a few hundred positions.

Given a subset $I$, the program at a high level executes the following steps.

1. Construct an imperfect-information subgame $\Gamma$ incorporating the saved computation from the previous move ($\hat{\Gamma}$), as well as the positions in the sampled subset $I$.

2. Compute an (approximately) optimal strategy profile (*i.e.*, an approximate Nash equilibrium) of $\Gamma$.

3. Use the Nash equilibrium to expand the game tree $\Gamma$.

4. Repeat the above two steps until a time budget is exceeded.

5. Select a move.

We now elaborate on each step individually. Full detail about our techniques, including formal descriptions of all techniques, proofs, and comparisions to prior work, can be found in Appendix C.

## 3.1 STEP 1: GENERATING THE INITIAL GAME TREE AT THE BEGINNING OF A TURN

The imperfect-information subgame $\Gamma$ is constructed from the old game tree $\hat{\Gamma}$ and the sampled additional positions $s \in I$ according to a new algorithm which we call *knowledge-limited unfrozen subgame solving* (KLUSS). It is more effective than the *knowledge-limited subgame solving* (KLSS) algorithm of Zhang and Sandholm (Zhang & Sandholm, 2021) (a comparison is presented in Appendix C). At a high level, KLSS and KLUSS address the issue of reasoning about common knowledge by assuming that sufficiently high-order knowledge is essentially irrelevant to game play: if there is a position $s$ in the old tree $\hat{\Gamma}$ such that we know that the opponent knows that we know that $s$ is not the true state, we remove $s$ from $\Gamma$ as it is assumed to be irrelevant. As an example, consider the game in Fig. 2. There are two players, ▲ and ▼. Suppose that we are ▲, and we have arrived at the circled node (which is alone in its infoset, *i.e.*, at this node, ▲ has perfect information).

The infosets (dotted lines) define a *connectivity graph* $G$ among the five nodes in that layer of the tree: two nodes $u$ and $v$ are connected if there is an infoset connecting any descendant of $u$ (including $u$ itself) to any descendant of $v$ (including $v$ itself). The nodes in that layer are labeled according to their distance from the circled node; the node labeled $\infty$ is not connected. Distance corresponds to order of knowledge: if the true node is the circled node, then the distance is the smallest integer $k$ for which the statement

$$\underbrace{\text{everyone knows that everyone knows that ... everyone knows that}}_{k \text{ repetitions}} \text{the true node is not } u$$

---

[6]Specifically, *Pluribus* (Brown & Sandholm, 2019) would not have been feasible without these poker-specific optimizations.

[7]A position describes where pieces are as well as the castling and *en passant* rights.

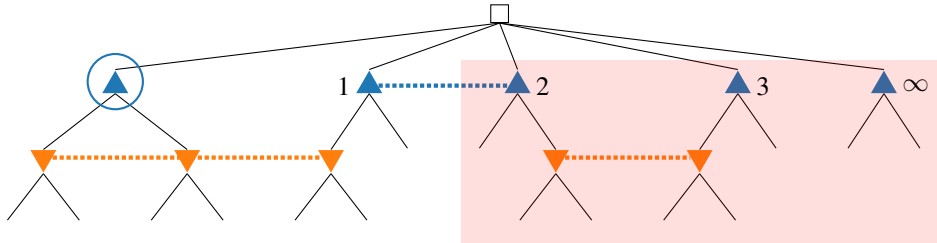

Figure 2: An example game tree, to illustrate KLUSS. The box (□) is a chance node. Dotted lines connect nodes in the same infoset.

is false. Thus the shaded red region corresponds to nodes that will be removed: everyone knows that these nodes are not the true nodes. This allows the game tree to be kept to a manageable size, even when the common-knowledge set (which the program never computes or uses) is large.

Our approach has two important properties. First, it enables the agent to reason about the opponent's information in a more powerful way than assuming something pessimistic, such as the opponent having perfect information (Parker et al., 2005; Russell & Wolfe, 2005). This allows behavior such as bluffing, which is important to strong play. Second, it accomplishes this while essentially only examining states $s$ that are relevant in the sense that, as far as our agent knows, the opponent might believe that $s$ is the true state. This ensures that, even when the common-knowledge set is large and poorly structured (*e.g.*, even when the vast majority of states in the common-knowledge set are irrelevant), subgame solving is still possible and effective.

The difference between KLUSS and KLSS is that in KLSS, the strategy at the two-node infoset for ▲ in Fig. 2 (and more generally all ▲-nodes at distance 1 and their descendants) is frozen to that node's strategy from the previous move. In KLUSS, it is unfrozen and will be game-theoretically optimized together with the rest of the subgame that is not removed (*i.e.*, not red).

KLSS and KLUSS are not always game-theoretically sound in theory because some of the removed (red in the figure) part of the game tree could be relevant to the decision, but they are often sound in practice (Zhang & Sandholm, 2021; Liu et al., 2023). They can be viewed as a computationally feasible alternative to traditional game-theoretically sound subgame solving.

## 3.2 STEP 2: EQUILIBRIUM COMPUTATION

The remaining steps are inspired by the *growing-tree counterfactual regret minimization* (GT-CFR) algorithm (Schmid et al., 2023): a game tree $\Gamma$ is simultaneously solved using an iterative equilibrium-finding algorithm and expanded using an expansion policy.

For equilibrium finding we use a state-of-the-art algorithm, *predictive CFR+* (PCFR+) (Farina et al., 2021). PCFR+ is an iterative, anytime algorithm for solving imperfect-information games, that can handle the fact that our game tree $\Gamma$ is changing over time. At all times $t$, PCFR+ maintains a profile $(x^t, y^t)$, where $x^t$ is our strategy and $y^t$ is the opponent's strategy.

PCFR+ has only been proven to converge *in average strategies*. That is, the empirical strategy profile $(\bar{x}^t, \bar{y}^t) := (\frac{1}{t}\sum_{s=1}^{t} x^s, \frac{1}{t}\sum_{s=1}^{T} y^s)$ converges to Nash equilibrium as $t \to \infty$. However, instead of computing the empirical average strategy, we circumvent this step and maintain only the last iterate $(x^t, y^t)$. There are several reasons for this choice, which are detailed in Appendix C.7.

## 3.3 STEP 3: EXPANDING THE GAME TREE

Nodes are selected for expansion by using carefully-designed *expansion policies* that balance exploration and exploitation. Our program chooses a node to expand by the following process. Fix one player to be the *exploring player*. (The choice of which player is exploring alternates: on odd-numbered iterations, P1 is the exploring player; on even-numbered iterations, P2 is the exploring player.) For this exposition, we will take P1 to be the exploring player. The *non-exploring* player will play according to its current strategy as computed by PCFR+, in this case $y^t$. The *exploring* player will play a perturbed version $\tilde{x}^t$ of its current strategy $x^t$. The strategy $\tilde{x}^t$ is designed to

balance between exploitation and exploration. *Exploitation* here means playing actions with high possible reward, that is, actions that have positive probability in $x^t$. *Exploration* means assigning positive probability to every possible action, to hedge against the possibility that the current tree incorrectly estimates the value of the action due to lacking search depth. For this, we use a method based on the *polynomial upper confidence bounds for trees* (PUCT) algorithm (Silver et al., 2016). Finally, a leaf node of the current tree $\Gamma$ is selected for expansion according to the strategy profile $(\tilde{x}^t, y^t)$.

One major difference between our algorithm and the GT-CFR algorithm lies in having only one player use the exploring strategy $\tilde{x}^t$, rather than both. Intuitively, this remains sound, because tree nodes that *neither* player plays to reach are irrelevant to equilibrium play, and thus do not need to be expanded. In Appendix C.4, we formally show that this variant, like GT-CFR, will find an exact equilibrium of any two-player zero-sum game given infinite search time. Thus, allowing one player to play directly from their equilibrium strategy (here, $y^t$) allows the tree expansion to be more focused. We call this GT-CFR variant *one-sided GT-CFR*.

Once a leaf node $z$ is chosen by the above process, its children are evaluated by a node heuristic and added to the game tree. The node heuristic is an estimate of the perfect-information value of $z$, as evaluated by the chess engine *Stockfish 14* (Stockfish). If $z$ is the first node in its infoset that has been expanded, a local regret minimizer is created for PCFR+, and it is initialized to pick the action with highest value according to the node heuristic. Theoretically, the guarantees of PCFR+ do not depend on the initialization, which can be arbitrary. However, practically, we find that initializing to a "good guess" of a good action leads to faster empirical convergence to equilibrium. More details can be found in Appendix C.5.

### 3.4 STEP 4: REPEAT

The above two steps are repeated, in parallel using a multi-threaded implementation, until a time budget is exceeded. Our implementation uses one thread running CFR and two threads expanding the game tree, which is shared across all three threads. The node expansion threads use locks to avoid expanding the same node, but the equilibrium computation thread uses no locks and only works on the already-expanded portion of the game tree. The time budget is set heuristically based on the amount of time remaining on the player's clock. Once the time budget is exceeded, the tree expansion threads (Step 3) are stopped first, and then, after a delay, the equilibrium computation thread (Step 2). The added time allocated to equilibrium computation is present so that a more precise equilibrium can be computed without the tree constantly changing.

### 3.5 STEP 5: SELECTING A MOVE

After those computations have stopped, a move is selected based on the (possibly mixed) strategy that PCFR+ has computed. Instead of directly sampling from this distribution, we first *purify* it (Ganzfried et al., 2012)—that is, we limit the amount of randomness. In particular, we sample from only the $m$ highest-probability actions, where $1 \leq m \leq 3$ is chosen based on the computed strategies. We only allow mixing ($m > 1$) when the algorithm believes that its computed strategy is *safe*—intuitively, this is when the algorithm's final strategy $x^T$ can guarantee expected value at least as good as what the algorithm thought to be possible before the turn. This purification technique made a significant difference in practice, detailed via an ablation test in Section 4.1.

## 4 EXPERIMENTAL EVALUATION

To evaluate our techniques, we conducted several experiments. The first was a 1,000-game match against the previous state-of-the-art AI for FoW chess (Zhang & Sandholm, 2021) (hereafter *ZS21*). Our new AI scored 85.1% (+834 =33 -133)[8], confidently establishing its superiority.

We then ran two experiments against human players. The first of these was a series of games against human players of varying skill levels. *Obscuro* played a total of 117 games (with time control 3 minutes + 2 seconds per move). This time control was selected because it was the most popular time

---

[8]This notation means 834 wins, 33 draws, and 133 losses.

control played on the most popular website for FoW chess (chess.com) at the time of the experiment. While in regular chess both fast and slow games are common, in FoW chess slow games are typically not played. The skill levels of the players, measured by their chess.com Fog of War chess ratings, ranged from 1450 to 2006. We excluded 17 of the games for various reasons such as disconnections, the opponent leaving before the game finished, or the opponent clearly losing on purpose, leaving 100 completed games. *Obscuro* scored 97% (+97 =0 -3), establishing conclusively that it is stronger than humans of this level.

Finally, we invited the top FoW chess player to a 20-game match (again at 3+2 time control). At the time of our match, *i.e.*, as of the rating list on August 16, 2024 (Chess.com, 2024), this player was rated 2318 and ranked #1 on the chess.com Fog of War blitz leaderboard. The games were played over the course of two days, 10 games per day, giving the human player an opportunity to analyze the first set of games overnight. In this match, *Obscuro* scored 80% (+16 =0 -4, +241 Elo), a conclusive and statistically significant ($p = 0.0118$ using an exact binomial test) victory against the world's strongest player. We thus conclude that *Obscuro* is superhuman.

The 20 games played against the top human are available through the following link: `https://lichess.org/study/sja93Uc0`

A curated sample of particularly interesting games from our 100 games played against humans of varying skill levels, including all three games lost by *Obscuro*, is available through the following link: `https://lichess.org/study/1zHFym7e`

In both links, each game lists which side *Obscuro* played (Black or White) and the game result (Win or Loss). All games are shown from the perspective of *Obscuro*.

## 4.1 ABLATIONS

In addition, we conducted multiple ablations with *Obscuro*. In each of these experiments, we turned off one or more of the new techniques introduced in this paper in order to evaluate the contributions of the different techniques to the performance of *Obscuro*. All ablations were run at a time control of 5 seconds per move. Unless otherwise stated, all ablations were *Obscuro* playing against a version of *Obscuro* with the single stated technique turned off. Recall from above that *Obscuro* with all techniques turned on scored 85.1% against ZS21 and 80% against the top human.

1. *Purification off.* This version allowed mixing among all stable actions, even if the current margin is negative or there are more than three of them.

   In a 1,000-game match, *Obscuro* scored 70.2% (+662 =79 -259).

2. *KLUSS off.* In this version, the strategies in infosets not touching our infoset were frozen, as in 1-KLSS.

   In a 1,000-game match, *Obscuro* scored 58.0% (+532 =96 -372).

3. *One-sided GT-CFR off.* In this version we use the two-sided node expansion algorithm proposed by the original GT-CFR paper (Schmid et al., 2023).

   In a 10,000-game match, *Obscuro* scored 53.3% (+4535 =1583 -3882).

4. *Non-uniform Resolve distribution off.* In Appendix C.3, we describe a modification to the *Resolve* subgame solving (Burch et al., 2014) algorithm that we made for *Obscuro*, in which the root nodes of the game tree are weighted using a nonuniform distribution, instead of the uniform distribution prescribed by Burch et al. (2014). For this ablation, we turned this modification off, and instead used the uniform distribution, as was done in prior papers on subgame solving including ZS21.

   In a 10,000-game match, *Obscuro* scored 53.3% (+4595 =1478 -3927).

5. *Two-sided GT-CFR only, against ZS21.* In this ablation, we turned off all the above improvements 1, 2, 3, and 4, and matched the resulting agent against that of ZS21. This serves to isolate the effect of using GT-CFR compared to using the LP-based equilibrium computation and iterative deepening node expansion as in ZS21.

   In a 1,000-game match, the two-sided GT-CFR version scored 72.6% (+711 =30 -259) against ZS21.

6. *Weaker evaluation function.* To test the impact of the evaluation function, we hand-crafted a simple evaluation function that takes into account only the material difference and number of squares visible to each player. We substituted this evaluation function in place of *Stockfish 14*'s neural network-based evaluation function, creating a new agent that we call *simple-eval (SE) Obscuro*. This evaluation function is very simple, and would not be well suited to regular chess. We tested *simple-eval Obscuro* against both *Obscuro* and ZS21.

   In a 1,000-game match, *Obscuro* scored 81.9% (+787 =63 -150) against *SE Obscuro*.

   In a 10,000-game match, *SE Obscuro* scored 55.0% (+5258 =486 -4256) against ZS21.

   This experiment shows that the evaluation function has a significant impact on the performance of *Obscuro*. Yet, the search algorithm is also vital: even a simplistic evaluation function with our improved search techniques is enough to be superior to ZS21.

All results in this and the next subsection are highly statistically significant ($z > 5$). The results suggest that each improvement played a significant role in the improvement of *Obscuro* over the previous state-of-the-art AI.

## 4.2 OTHER EXPERIMENTS

Finally, to test the effect of the time limit on the performance of *Obscuro*, we tested versions of *Obscuro* with different time limits against each other. The results were as follows. All matches consisted of 10,000 games.

- *Obscuro* with $\frac{1}{8}$s/move scored 56.4% (+5162 =943 -3895) against *Obscuro* with $\frac{1}{16}$s/move.
- *Obscuro* with $\frac{1}{4}$s/move scored 56.5% (+5031 =1231 -3738) against *Obscuro* with $\frac{1}{8}$s/move.
- *Obscuro* with $\frac{1}{2}$s/move scored 56.7% (+4923 =1503 -3574) against *Obscuro* with $\frac{1}{4}$s/move.
- *Obscuro* with 1s/move scored 54.0% (+4617 =1566 -3817) against *Obscuro* with $\frac{1}{2}$s/move.
- *Obscuro* with 2s/move scored 53.7% (+4589 =1561 -3850) against *Obscuro* with 1s/move.
- *Obscuro* with 4s/move scored 52.3% (+4463 =1530 -4007) against *Obscuro* with 2s/move.
- *Obscuro* with 8s/move scored 52.4% (+4501 =1482 -4017) against *Obscuro* with 4s/move.
- *Obscuro* with 16s/move scored 52.3% (+4448 =1563 -3989) against *Obscuro* with 8s/move.

These results, converted to the standard Elo scale, are visualized in Fig. 3. As expected and in line with known results for other settings (*e.g.*, for regular chess (Silver et al., 2017)), increasing search time has a significant impact on playing strength, but with somewhat diminishing returns.

Finally as a sanity check, we also tested *Obscuro* against a random opponent. The only realistic way for *Obscuro* to lose to a random opponent is by not defending against **Qa4+** or **Qa5+** in the opening as previously discussed, which happens with only very small probability. As previously discussed, occasionally losing to a weak (here, random) player would not in itself evidence that *Obscuro* is playing suboptimally, since even an exact equilibrium player should lose to a random player with positive probability. Nonetheless, *Obscuro* won 1000 consecutive games against the random opponent.

## 5 CONCLUSIONS AND FUTURE RESEARCH

We presented the first superhuman agent for FoW chess, *Obscuro*. Our agent is completely based on real-time search. Thus, *Obscuro* ran on regular consumer hardware, in contrast to most prior superhuman efforts involving search that we have discussed, which have run on large computing clusters with far more computing power at play time. This demonstrates the power of search alone. FoW chess is now the largest (measured by amount of imperfect information) turn-based game in which superhuman performance has been achieved and the largest game in which imperfect-information search techniques have been successfully applied.

Since FoW chess is somewhat similar to regular chess, it was sufficient to combine a perfect-information evaluation function from regular chess (namely, that used by *Stockfish*) with our game-independent state-of-the-art search algorithms for imperfect-information games. Also, *Obscuro*

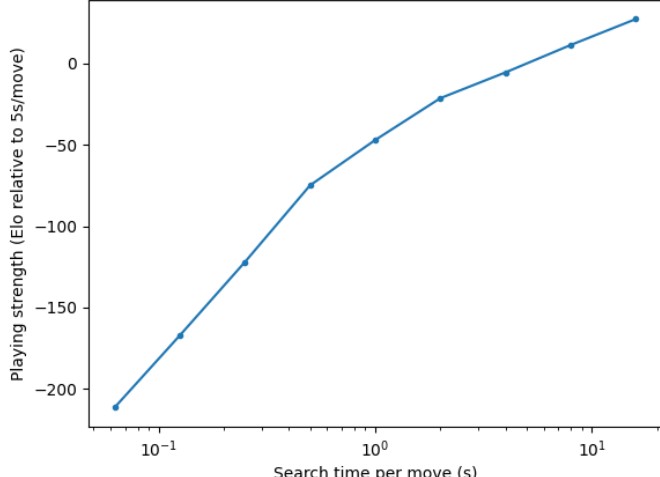

Figure 3: Visualization of time scaling of *Obscuro*. The $y$-axis is relative to the playing strength of *Obscuro* with 5 seconds per move.

stores at all times the entire set of possible states in memory. While these techniques were feasible for FoW chess—due to the similarity to regular chess and the relatively small infosets—one can imagine even more complex games on which they will not work directly.

Even more complex settings could be tackled by merging our techniques with deep reinforcement learning to learn the evaluation function, instead of using a perfect-information-game evaluation function (in our case, from *Stockfish*), and/or using *continuation strategies* (Brown & Sandholm, 2019) to mitigate game-theoretic issues caused by using node-based evaluation functions in imperfect-information games. In a different direction, further play strength and scalability could be achieved by sampling from an infoset using a model of opponent behavior instead of doing so uniformly.

## ACKNOWLEDGMENTS

We thank Stephen McAleer for helpful discussions. This material is based on work supported by the Vannevar Bush Faculty Fellowship ONR N00014-23-1-2876, National Science Foundation grants RI-2312342 and RI-1901403, ARO award W911NF2210266, and NIH award A240108S001. B.H.Z. was also supported by the CMU Hans J. Berliner Graduate Fellowship in Artificial Intelligence.

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

## A    Rules of FoW chess

FoW chess is identical to regular chess, except for the following differences (Chess.com, 2025b).

- A player wins by capturing the opposing king. There is no check or checkmate. Thus:
    - Moving into (or failing to escape) a check is legal and thus results in immediate loss.
    - Castling into, out of, or through check is legal (though, of course, castling into check loses immediately).
    - Stalemate is a forced win for the stalemating player.
    - There is no draw by insufficient material. In particular, KN vs K is a strong position for the KN, and even K vs K is not an immediate draw (although K vs K is drawn in equilibrium except in some literal edge cases where one king is on the edge of the board and cannot immediately escape.)

- After every move, each player observes all squares onto which her pieces can legally move.

- If a pawn is blocked from moving forward by an opposing piece (or pawn), the square on which the opposing piece/pawn sits is *not* observed. Thus, the player knows that the pawn is blocked, but not what is blocking it (unless, of course, some other piece can capture it.)

- If a pawn can capture *en passant*, the pawn that can be captured *en passant* is visible.

    In particular, the above rules imply that both players always know their exact set of legal moves.

- Threefold repetition and 50-move-rule draws do not need to be claimed. In particular, a draw under either rule can happen without either player knowing for certain until it happens and the game ends.

## B    Notation and Preliminaries

The techniques used in *Obscuro* are general, so in this section we will formulate them in terms of general extensive-form games. To do this, we need to introduce some notation.

### B.1    Notation and Preliminaries

A *two-player zero-sum timeable extensive-form game* (hereafter simply *game*) consists of:

1. a tree of *histories* $H$, rooted at the *empty history* $\varnothing \in H$. The set of leaves (*terminal nodes*) of $H$ is denoted $Z$. Each downward edge out of a non-leaf node $h \in H \setminus Z$ is labeled with a distinct *action* or *move*. The node reached by following the edge (action) $a$ at node $h$ is denoted $ha$. The set of actions available at $h$ is denoted $A(h)$.

2. a *payoff function* $u : Z \to [-1, +1]$,

3. a partition $H \setminus Z = H_{\mathsf{C}} \sqcup H_{\blacktriangle} \sqcup H_{\blacktriangledown}$, denoting whose turn it is—that is, for each $i \in \{\mathsf{C}, \blacktriangle, \blacktriangledown\}$, $P_i$ is the set of nodes at which player $i$ moves. Player $\mathsf{C}$ is chance, who plays according to a fixed strategy $p(\cdot|h)$.[9]

4. for each player $i \in \{\blacktriangle, \blacktriangledown\}$ and each $h \in H$, an *observation* $o_i(h)$ that player $i$ receives upon reaching $h$. The observation uniquely determines whether $h \in P_i$ (*i.e.*, whether it is player $i$'s turn) and, if it is, the set of legal actions. That is, if $o_i(h) = o_i(h')$, then $h \in P_i$ if and only if $h' \in P_i$, and if so, $A(h) = A(h')$.

We will use $\preceq$ to denote the precedence relation on a tree. For example, if $h, h'$ are histories then $h \preceq h'$ means $h$ is an ancestor of $h'$. If $s, s'$ are sequences of player $i$, then $s \preceq s'$ means that $s$ is a prefix of $s'$. If $S$ is a set, $s \succeq S$ means $s \succeq s'$ for some $s' \in S$. The *downward closure* of $S$ is $\bar{S} := \{h : h \succeq S\}$.[10]

---

[9]FoW chess contains no chance moves, but we include this in the interest of generality.

[10]In this paper, we visualize trees expanding from the top downwards, so $\bar{S}$ is the set of descendants of $S$.

We will distinguish between *states* and *histories*. A *state* is a sufficient statistic for future play of the game. That is, all data about the subtree rooted at a history $h$ is uniquely determined by the state at $h$. Multiple histories can have the same state.

The *sequence* of a player $i$ upon reaching a node $h \in H$ is the sequence of observations made and actions played by $i$ so far. Two nodes $h, h'$ are *indistinguishable to player $i$*, written $h \sim_i h'$, if they have the same sequence for player $i$. An equivalence class of $H$ under $\sim_i$ is an *infoset*, for player $i$. Throughout this paper, $I$ will denote a ▲-infoset, and $J$ will denote a ▼-infoset. We will assume, without loss of generality, that the player sequence and opponent sequence together uniquely specify a game tree node—that is, $|I \cap J| = 1$ for every ▲-infoset $I$ and ▼-infoset $J$.

By convention, information sets containing nodes at which player $i$ is not the acting player are typically not drawn (and often not even defined); in our paper, we will need them in order to define the knowledge graph. Thus let $\mathcal{I}$ (resp. $\mathcal{J}$) denote the set of infosets at which ▲ (resp. ▼) is the acting player. If $a$ is a legal action at an infoset $I \in \mathcal{I}$, the sequence reached by playing $a$ at $I$ is $(I, a)$. The set of nodes $\{ha : h \in I\}$ will also be denoted $(I, a)$. Let $s_i(h)$ denote the sequence of player $i$ at $h$, as of the last time player $i$ played an action. Thus, $s_{\blacktriangle}(h)$ (resp. $s_{\blacktriangledown}(h)$) can be identified with a pair $(I, a)$ where $I \in \mathcal{I}$ (resp. $(J, a)$ where $J \in \mathcal{J}$).

A *(behavioral) strategy* of ▲ (resp. ▼) is a selection of a distribution of actions at each infoset, $x \in X = \bigtimes_{I \in \mathcal{I}} \Delta(A(I))$ (resp. $y \in Y = \bigtimes_{J \in \mathcal{J}} \Delta(A(J))$). We will use the general notation $x(u'|u)$, where $u \preceq u'$ to denote the probability that ▲ plays *all* actions on the $u \to u'$ path, where $u$ and $u'$ are sequences, infosets, or nodes. Similarly, $x(a|u)$ denotes the probability that $x$ takes action $a$ at $u$ (when $u \in \mathcal{I}$ or $u \in H_{\blacktriangle}$). If the right half is omitted, *e.g.*, $x(u)$, it is understood to be $\varnothing$, *e.g.*, $x(u) = x(u|\varnothing)$. In particular, $x(h)$ denotes the probability that ▲ plays all actions on the $\varnothing \to h$ path. Similar notation is used for ▼.

The expected value for ▲ in strategy profile $\pi = (x, y)$ is $u(\pi) := \mathbb{E}_{z \sim \pi} u(z)$ where the expectation is over terminal nodes $z$ when ▲ plays $x$ and ▼ plays $y$. (Since the game is zero sum, the value for ▼ is $-u(\pi)$.)

The *conditional value* $u(\pi|S)$ is the conditional expectation given that some node in the set $S$ is hit. The *(conditional) best-response value* $u^*(x|J, a)$ to a ▲-strategy $x \in X$ upon playing action $a$ at infoset $J$ is the best possible conditional value that ▼ against $x$ after playing $a$ at $J$:

$$u^*(x|J, a) = \min_{y \in Y : y(J, a) = 1} u(x, y|J).$$

*Counterfactual values* (CFVs), which we will denote by $u^{\mathrm{cf}}$, are defined similarly to conditional values, but scaled by the probability of the other players playing to $J$:

$$u_{\blacktriangle}^{\mathrm{cf}}(x, y; J, a) = u(x, y|J, a) \cdot \sum_{h \in J} x(h)p(h) = \sum_{z \succeq J} x(z)p(z)y(z|h)u(z).$$

The best-response value at infoset $J$ is $u^*(x|J) = \min_a u^*(x|J, a)$. The best-response value $u^*(x)$ is $\min_{y \in Y} u(x, y) = u^*(x|\varnothing)$. Analogous definitions hold when the players are swapped.

## B.2 Order-$k$ knowledge, common knowledge, and subgame solving

In this section, we present the mathematical notation we will use in the rest of the appendix, and relevant prior work on subgame solving (see, *e.g.*, (Kovařík et al., 2021) for an overview).

The *connectivity graph* $G$ of a game $\Gamma$ is the graph whose vertices are the nodes of $\Gamma$, and whose edges connect nodes in the same infoset of any player. Now let $I$ be any infoset.[11] The *order-$k$ knowledge set* $I^k$ is the set of nodes at distance strictly less than $k$ from some node in $I$. (In particular, $I^1 = I$.) The *common-knowledge set* $I^\infty$ is the set of all nodes a finite distance away from some node in $I$, *i.e.*, it is the connected component in $G$ containing $I$. Intuitively, the distance from $I$ captures the level of common knowledge. If the true node is $h \in I$, then

    1. ▲ knows $h \in I$,

---

[11] The definition of $I^k$ can also be applied to arbitrary sets of nodes $I$, but here we will only need it for infosets.

2. $\blacktriangle$ knows $\blacktriangledown$ knows $h \in I^2$,

3. $\blacktriangle$ knows $\blacktriangledown$ knows $\blacktriangle$ knows $h \in I^3$,

and so on. Hence the statement $h \in I^\infty$ is common knowledge.

In the remainder of the paper, we take the perspective of the maximizing player $\blacktriangle$. Subgame solving starts with a *blueprint strategy profile* $(x, y)$. In *Obscuro*, the blueprint strategy profile is simply the saved strategy from the computation on the previous move; on the first move, subgame solving does not require a blueprint.

Suppose that we reach infoset $I$ during a game. Before selecting a move at $I$, we would like to do some computation to compute a new strategy $x'$ that we will use instead of $x$. That is, we would like to perform *subgame solving*.

We will first describe two common variants of *common-knowledge* subgame solving: *Resolve* (Burch et al., 2014) and *Maxmargin* (Moravcik et al., 2016), both of which we will use in *Obscuro*. Both variants begin by constructing a *gadget game* using common-knowledge set $I^\infty$, and are based on the principle of searching for a strategy $x'$ that does not worsen the opponent's best response values. More formally, let $M(x', J) := u^*(x'|J) - u^*(x|J)$ be the *margin* at $\blacktriangledown$-infoset $J \subseteq I^\infty$. The *alternate value* $u^*(x|J)$ is the value to which $\blacktriangle$ must restrict $\blacktriangledown$ at $J$ in order to ensure that exploitability does not increase.

*Maxmargin* and *Resolve* differ in how they aggregate the margins across the different information sets $J \in \mathcal{J}_0$. The *Maxmargin* objective is to maximize the minimum margin:

$$\max_{x'} \min_{J \in \mathcal{J}_0} M(x', J).$$

The *Resolve* objective is to maximize the average margin truncated to zero:

$$\max_{x'} \frac{1}{|\mathcal{J}_0|} \sum_{J \in \mathcal{J}_0} [M(x', J)]^- \quad \text{where} \quad [z]^- := \min\{0, z\}.$$

and $\mathcal{J}_0 := \{J : J \subseteq I^\infty\}$ is the set of possible root infosets for $\blacktriangledown$ in the subgame.

A subgame solving method is *safe* if applying it cannot increase exploitability of the overall agent compared to not applying it—*i.e.*, compared to playing the blueprint strategy. Both *Maxmargin* and *Resolve* are safe, assuming of course that subgames are solved exactly.[12]

Subgame solving via *Resolve* and *Maxmargin* can also be performed using *gadget games*. In *Resolve*, the following gadget game is played. First, chance chooses a node $h \in I^\infty$ with probability proportional to $p(h)x(h)$. Then, $\blacktriangledown$ observes the infoset $J \ni h$, and decides whether to *play* or *exit*. If $\blacktriangledown$ exits, the game ends immediately with utility equal to the alternate value $u^*(x|J)$. Otherwise, the game continues as normal from node $h$. In *Maxmargin*, $\blacktriangledown$ first selects the infoset $J \in \mathcal{J}_0$, and then chance samples a node $h \in J$ with probability proportional to $p(h)x(h)$. Then, $\blacktriangle$ immediately receives utility $-u^*(x|J)$.[13] Chance then selects a node $h \in J$ with probability proportional to $p(h)x(h)$, and the game continues from $h$.

*Maxmargin* and *Resolve* have very different behavior. When it is impossible to make all margins nonnegative (due to approximations), *Maxmargin* will make the *pessimistic* assumption that the opponent will play the worst infoset, whereas *Resolve* will, roughly speaking, assume that the opponent will play uniformly over all infosets with negative margin. On the other hand, when it is possible to make all margins nonnegative, there is a set of subgame strategies that are maximizers of the *Resolve* objective, that is, equilibria of the *Resolve* gadget game. *Resolve* allows any one of these strategies to be selected, whereas *Maxmargin* enforces that the strategy be in particular the one that maximizes the minimum margin.

In the state-of-the-art common knowledge subgame-solving technique, *reach subgame solving* (Brown & Sandholm, 2017), any gifts given to us by the opponent through mistakes in reaching the subgame can be given back to the opponent within the subgame; this enlarges the strategy space

---

[12]In our application, safety is hard to reason about: neither the blueprint strategy $x$ nor the subgame-solved strategy $x'$ are full-game strategies, so asking the question of which is less exploitable is strange.

[13]This can be implemented, for example, by adding $u^*(x|J)$ to the value of every terminal node $z \succeq J$ in the subgame.

that we can optimize over safely and thus has been shown to yield stronger play (*e.g.*, in poker games (Brown & Sandholm, 2017; 2018; Brown et al., 2018; Brown & Sandholm, 2019)). This is done by adjusting the alternate values $u^*(x|J)$ in the case when ▼ provably made a mistake(s) in playing to reach $J$. Reach subgame solving can be applied on top of either *Resolve* or *Maxmargin*. In particular, the value

$$g(J) := \sum_{J'a' \preceq J} \left[ u_{\blacktriangledown}^{\text{cf}*}(x; J'a') - u_{\blacktriangledown}^{\text{cf}*}(x; J') \right],$$

which is an estimate of the gift using the current strategy profile $(x, y)$, is added to the alternate value at each infoset $J \in \mathcal{J}_0$.

In those prior subgame-solving techniques and ours, the desired gadget game then replaces the full game, and its solution is used to select a move at $I$. When a new infoset is reached, the process repeats, with the solution to the previous subgame taking the place of the blueprint.

## C  FURTHER DETAILS ABOUT *Obscuro*

We now give more details about *Obscuro*. Appendix C.1 details the techniques that are identical to the prior SOTA (ZS21). The remaining subsections detail the improvements over ZS21 that we developed in *Obscuro*. We also include pseudocode for the major components of *Obscuro*, in Figures 8-12.

### C.1  PRIOR STATE OF THE ART IN FOW CHESS

*Obscuro* decides between *Maxmargin* and *Resolve* by examining the current objective value in the subgame. If ▼ always chooses to exit in the resolve gadget (*i.e.*, the current strategy is *safe*), *Maxmargin* is used. Otherwise, *Resolve* is used. This switch may happen, even multiple times, in the middle of the search process for a move, if the subgame value is fluctuating. Intuitively, this choice prevents the agent from being too pessimistic when faced with novel situations that it did not anticipate.

Between moves, *Obscuro* maintains the list of all possible states given its current sequence of observations, as well as the search tree and current approximate equilibrium strategy profile $(x, y)$ from the previous search. This previous strategy profile $(x, y)$ is used as the blueprint for subgame solving.

When it is *Obscuro*'s turn, *Obscuro* first builds both the *Maxmargin* and *Resolve* gadget subgames. The gadget subgames share the same game tree in memory after the subgame root layers. Thus, for example, node expansions and strategy updates for infosets beyond the subgame root layers apply to both subgame gadgets. This allows the transition between the two subgames, if necessary, to be smoothly executed.

If insufficiently many nodes exist in the sample of *Obscuro*'s current infoset $I$, nodes are added by sampling at random without replacement from the set of possible states. At newly-added nodes $h$, the opponent is assumed to have perfect information, and the alternate value is set to $\min\{\tilde{v}(h), v^*\}$ where $v^*$ was the expected value of *Obscuro* in the previous search, and $\tilde{v}(h)$ is *Stockfish*'s evaluation function evaluated at $h$.

### C.2  BETTER ALTERNATE VALUES AND GIFT VALUES

For alternate values in both *Resolve* and *Maxmargin*, in *Obscuro* we use $u(x, y|J)$ instead of the best-response value $u^*(x|J)$ which is more typically used in subgame solving as we described in Appendix B.2 (Brown & Sandholm, 2017). Similarly, we use the counterfactual values $u^{\text{cf}}(x, y; J, a)$ and $u^{\text{cf}}(x, y; J)$ to define the gift instead of the counterfactual best responses $u^{\text{cf}*}(x; J, a)$ and $u^{\text{cf}*}(x; J)$, resulting in the gift estimate

$$\hat{g}(J) := \sum_{J'a' \preceq J} [u^{\text{cf}}(x, y; J'a') - u^{\text{cf}}(x, y; J')]^+$$

These changes are for stability reasons: especially late in the tree, the current strategy $x$ may be inaccurate, and the best-response value $u^*(x|J)$ may not be an accurate reflection of the quality

of the blueprint strategy $x$, especially near the top of the tree. Of course, if $(x, y)$ is actually an equilibrium of the constructed subgame, then these values are the same.

### C.3 BETTER ROOT DISTRIBUTION FOR *Resolve*

When using *Resolve*[14] for the subgame solve in KLSS in games with no chance actions, the standard algorithm for *Resolve* will choose an opponent infoset $J$ uniformly at random from the distribution of possible infosets. In reality, the correctness of *Resolve* does not depend on the distribution chosen, so long as it is fully mixed. To be more optimistic, we therefore use a different distribution. We choose an infoset $J$ via an even mixture of a uniformly random distribution and the distribution of infosets generated from the opponent strategy in the blueprint. That is, the probability of the subgame root being infoset $J$ is

$$\alpha(J) := \frac{1}{2} \left( \frac{y(J)}{\sum_{J'} y(J')} + \frac{1}{m} \right),$$

where $m$ is the number of ▼-infosets in the current subgame and the sum is taken over those same infosets. In other words, the *Resolve* objective becomes

$$\max_{x'} \sum_{J \in \mathcal{J}_0} \alpha(J)[M(x', J)]^-.$$

In this manner, more weight is given to those positions that were found to be likely in the previous iteration, while maintaining at least some positive weight on every strategy.

### C.4 BETTER NODE EXPANSION VIA GT-CFR

*Growing-tree CFR* (GT-CFR) (Schmid et al., 2023) is a general technique for computing good strategies in games. Intuitively, it works, like PUCT, by maintaining a current game $\tilde{\Gamma}$ and simultaneously executing two subroutines: one that attempts to solve the game $\tilde{\Gamma}$, and one that expands leaf nodes of $\tilde{\Gamma}$. As mentioned in the body, we use PCFR+ for game solving.

For expansion, we use a new variant of GT-CFR which we call *one-sided GT-CFR*, which, unlike PUCT and GT-CFR, may only expand a small fraction of nodes in the tree. As stated in the body, our one-sided GT-CFR algorithm selects the node to expand according to the profile $(\tilde{x}^t, y^t)$, where $y^t$ is the *non-expanding player's current CFR strategy* and $\tilde{x}^t$ is an exploration profile constructed from the expanding player's current strategy.[15] As in GT-CFR, the expanding player's strategy $\tilde{x}^t$ is a mixture of a strategy $\tilde{x}^t_{\text{Max}}(a|I)$ derived from the player's current strategy $x^t$ and an exploration strategy $\tilde{x}^t_{\text{PUCT}}(a|I)$ derived from PUCT (Silver et al., 2016). In particular, we define

$$\tilde{x}^t_{\text{Max}}(a|I) \propto \mathbf{1}\{x^t(a|I) > 0\}$$

to be the uniform distribution over the support of the current CFR strategy, and

$$\tilde{x}^t_{\text{PUCT}}(a|I) = \mathbf{1}\{a = \operatorname*{argmax}_{a'} \bar{Q}(I, a)\}$$

where

$$\bar{Q}(I, a) = u(x^t, y^t | I, a) + C\sigma^t(I, a)\frac{\sqrt{N^t(I)}}{1 + N^t(I, a)}.$$

Here, $C$ is a tuneable parameter (which we set to 1); $\sigma^t(I, a)$ is the empirical variance of $u(x^t, y^t | I, a)$ over the previous times we have visited $I$ during expansion (with two prior samples of $-1$ and $+1$ to ensure it is never zero); $N^t(I)$ is the number of times infoset $I$ has been visited during expansion; and $N^t(I, a)$ is the number of times action $a$ has been selected. Finally, as in GT-CFR, we define

$$\tilde{x}^t_{\text{sample}}(a|I) = \frac{1}{2}\tilde{x}^t_{\text{Max}}(a|I) + \frac{1}{2}\tilde{x}^t_{\text{PUCT}}(a|I).$$

Unlike GT-CFR as originally described (Schmid et al., 2023), our one-sided GT-CFR works on the *game tree itself*, not the *public tree*. The public tree in our setting would be difficult to work with since the amount of common knowledge is very low.

---

[14]For *Maxmargin*, there is no prior distribution because the adversary picks the distribution.

[15]In this presentation, ▲ is the expanding player. When ▼ is the expanding player, the roles of $x$ and $y$ are also flipped. As stated in the body, the expanding player alternates between ▲ and ▼ after every node expansion.

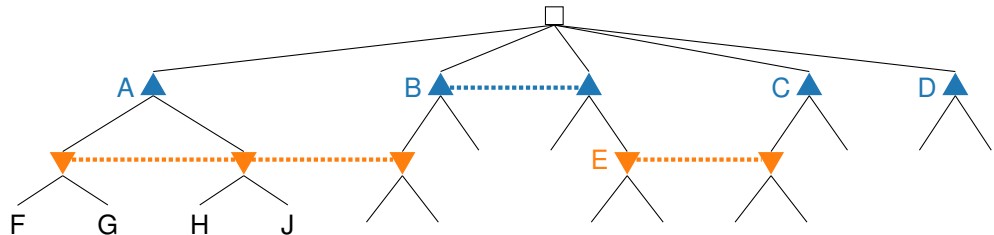

Figure 4: The game tree from Fig. 2, now with some nodes labeled, which will be referenced in the text.

Our one-sided GT-CFR, unlike PUCT and GT-CFR (Kocsis & Szepesvári, 2006; Schmid et al., 2023), is *not* guaranteed to eventually expand the whole game tree. For example, suppose that our game $\tilde{\Gamma}$ is as in Fig. 4, and that both players are currently playing the strategy "always play left". Then Node F is reached by both players, nodes G and H are reached by only one of the two players (▲ and ▼ respectively), and Node J is reached by neither player. As such, *Node J will not be expanded*, and if the current strategy is an equilibrium, this can be proven without knowing the details of any subtree that may exist at J.

Nonetheless, we can still show an asymptotic convergence result:

**Theorem 1.** *For any given $\epsilon > 0$, the average strategy profile $(\bar{x}, \bar{y})$ in one-sided GT-CFR eventually converges to an $\epsilon$-Nash equilibrium of any finite two-player zero-sum $\Gamma$.*[16]

*Proof.* Since $\Gamma$ is finite, eventually one-sided GT-CFR stops expanding nodes. At this time, let $\tilde{\Gamma}$ be the expanded game tree. Since no more nodes are expanded, and CFR is correct, one-sided GT-CFR eventually converges to an approximate Nash equilibrium $(\bar{x}, \bar{y})$ of $\tilde{\Gamma}$. At this time, it is perhaps the case that there remain unexpanded nodes in the current tree $\tilde{\Gamma}$. However, any such nodes must have been played with asymptotic probability 0 by *both* players; otherwise, if (say) ▼ plays to an unexpanded node $h$ with asymptotically positive probability, then $h$ would have been expanded at some point when ▲ was the expander. Thus, best-response values in $\tilde{\Gamma}$ are the same as they are in $\Gamma$, and therefore $(\bar{x}, \bar{y})$ is also an approximate equilibrium in $\Gamma$. □

### C.5 EVALUATING NEW LEAVES

When a (non-terminal) leaf node $z$ of $\tilde{\Gamma}$ is selected, it is expanded. That is, all of its children are added to the tree. To assign utility values the children of $z$, we run the open-source engine *Stockfish 14* (Stockfish), in *MultiPV* mode, at depth 1 on node $z$, which gives evaluations for all children of $z$ in a single call,[17] and clamp its result to $[-1, +1]$ in the same manner as done by ZS21 (Zhang & Sandholm, 2021).

When the children of $z$ are added to the tree, $z$ becomes a nonterminal node and hence will be placed in an infoset. If $z$ is the first node of its infoset to be expanded in $\tilde{\Gamma}$, we also need to initialize a new regret minimizer to be used by PCFR+ at this new infoset. Doing so naively would cause a sort of instability: the evaluation of $z$ will be (approximately) equal to the largest evaluation of any child of $z$ (due to how regular perfect-information evaluation functions work), but PCFR+ normally would initialize its strategy uniformly at random. Thus, the evaluation of $z$ would suddenly change to being the *average* of the evaluations of the children of $z$, which could be very different from the maximum (for example, if the move at $z$ is essentially forced). To mitigate this instability, we exploit

---

[16]Technically, $(\bar{x}, \bar{y})$ is only a partial strategy in $\Gamma$, since it does not specify how to play after any unexpanded nodes. However, this is fine: *any* extension of $(\bar{x}, \bar{y})$ will be an equilibrium of $\Gamma$, and unexpanded nodes are not reached by either player. For clarity, as is typical for extensive-form games (see, *e.g.*, Zinkevich et al. (Zinkevich et al., 2007)), the average of strategies is always taken in sequence form. That is, $\bar{x}$ is the strategy for which $\bar{x}(h) = \frac{1}{T} \sum_{t=1}^{T} x^t(h)$.

[17]Using a single call has two minor advantages: first, it takes advantage of slight extensions that may be used in Stockfish at low depth; second, it reduces the overhead of calling Stockfish to one call per node being expanded, instead of one call per child of that node.

the property that, in CFR (and all its variants, including PCFR+), the first strategy can be arbitrary. Conventionally it is set to the uniform random strategy, but we instead set it by placing all weight on the best child of $z$ as evaluated by *Stockfish*.

### C.6 KNOWLEDGE-LIMITED SUBGAME SOLVING

ZS21 (Zhang & Sandholm, 2021) uses *knowledge-limited subgame solving* (KLSS). KLSS results from two changes to common-knowledge subgame solving. Let $I$ be the current infoset of ▲. As an example, consider the game in Fig. 4, and let $I$ be the infoset at A. Let $k$ be an odd positive integer. Then ZS21 (Zhang & Sandholm, 2021) defines $k$-KLSS by making the following two changes.

1. Nodes outside the downward closure $\overline{I^{k+1}}$ are completely removed from the game tree. In Fig. 4, this would amount to removing the subtrees rooted at C, D, and E.

2. ▲-nodes in $\overline{I^{k+1}} \setminus \overline{I^k}$ are frozen to their strategies in the blueprint, *i.e.*, they are made into chance nodes with fixed action probabilities. In Fig. 4, this amounts to making Node B a chance node.

ZS21 sets $k = 1$ in their FoW chess agent. Freezing the ▲-nodes in $\overline{I^2} \setminus \overline{I^1}$ allows their equilibrium-finding module, which is based on linear programming, to scale more efficiently, since the nodes in that subtree are now only dependent on ▼'s strategy, not ▲'s.

KLSS, as implemented by ZS21, already lacks safety guarantees: they have an explicit counterexample in which using KLSS may decrease the quality of the strategy relative to just using the blueprint. We make one simple change to KLSS for *Obscuro*: we allow ▲-nodes in $\overline{I^2} \setminus \overline{I}$ to be unfrozen and hence re-optimized in the subgame. We may call this 2-knowledge-limited *unfrozen* subgame solving (KLUSS),[18] since its complexity depends on the order-2 subgame $\overline{I^2}$. 2-KLUSS essentially amounts to pretending that $\overline{I^2} = \overline{I^\infty}$.

We now make a few remarks about KLUSS.

1. Like 1-KLSS, 2-KLUSS lacks safety guarantees in the worst case. However, KLSS is often safe in practice (Zhang & Sandholm, 2021), and KLUSS outperforms KLSS in FoW chess as we showed in the ablations in Section 4.1. There are two further considerations:

   (a) *Obscuro* does not *have* a full-game blueprint: its blueprint is simply the strategy from the previous timestep, which is depth limited. Thus, we *must* use some form of subgame solving to play the game. KL(U)SS is currently the only variation of subgame solving that is both somewhat game-theoretically motivated for imperfect-information games and computationally feasible in a game like FoW chess.

   (b) Although both KLSS and KLUSS are unsafe in the worst case, it should be heuristically intuitive that they should improve performance *more* when the blueprint itself is of low quality. Indeed, we *expect* our "blueprints" (strategies carried over from the previous timestep) to have rather low quality, especially deep in the search tree where such strategies are based on very low-depth search! So, we believe heuristically that using KL(U)SS in this manner should usually be game-theoretically sound.

2. Since our equilibrium-finding module for *Obscuro* is based on CFR instead of linear programming—in particular, it uses the full game tree $\tilde{\Gamma}$ instead of a sequence-form representation—it does not benefit from freezing the ▲-nodes in $\overline{I^2} \setminus \overline{I^1}$, since those nodes would still need to be maintained. Thus, there is less reason for us to freeze those nodes. Further, with straightforward pruning techniques (namely, *partial pruning* (Brown & Sandholm, 2015)), CFR iterations usually take *sublinear* time in the size of the game tree (unlike linear programming, which takes at least linear time in the representation size), reducing the need to optimize the size of the game representation.

3. Again since we use CFR, the solutions that are computed by the equilibrium-finding module are inherently *approximate*, and especially at levels deep in the tree, their approximation can be relatively poor. As such, allowing these infosets to be unfrozen gives them the chance to learn better actions.

---

[18]This can be easily generalized to $k$-KLUSS for any $k$.

4. 1-KLSS removes the nodes in $\overline{I^2} \setminus \overline{I^1}$, folding them into the sequence-form representation for efficiency. In contrast, our approach of maintaining these nodes allows them to be *selected for expansion*. This fixes a weakness of ZS21: ZS21 was only capable of searching for bluff opportunities "locally", since any ▲-node in $\overline{I^2} \setminus \overline{I^1}$ would cease to be in the tree once the search horizon was passed. In contrast, *Obscuro* is capable of maintaining ▲-nodes in $\overline{I^2} \setminus \overline{I^1}$ for a long time, allowing deeper bluff opportunities.

5. Liu et al. (2023) introduced a *safe* variant of KLSS, which they call *safe KLSS*, in which the subgame solver attempts to find a subgame strategy $x'$ that maintains at least the same value for every *opponent strategy* $y$, instead of against every *infoset* $J$. This is a much stricter condition that is much more difficult to satisfy and thus substantially constrains the strategy to be close to the blueprint. Therefore, the safety requirement significantly decreases the power and value of subgame solving, especially when the blueprint is bad. Moreover, safe KLSS drops all nodes outside $\overline{I^1}$, which once again introduces the problem of the previously-listed item: if we were to use safe KLSS in our setting, our AI would not be capable of exploiting long bluff opportunities.

## C.7 SELECTING AN ACTION

As mentioned in the body, *Obscuro* selects its action using the *last iterate* of PCFR+, rather than the average iterate which is known to converge to a Nash equilibrium. We do this for two reasons.

1. The stopping time of the algorithm, due to the inherent randomness of processor speeds, is already slightly randomized. Thus, stopping on the last iterate does not actually stop at the same timestep $T$ every time: it in effect mixes among the last few strategies. Thus, we do not need to actually randomize ourselves to gain the benefit of randomization.

2. PCFR+ is conjectured (*e.g.*, Farina et al. (2024)) to exhibit last-iterate convergence as well. Indeed, we measured the Nash gap of the last iterate $(x^T, y^T)$ (in the expanded game $\tilde{\Gamma}$), and the typical Nash gap was approximately equivalent to half a pawn—much less than the reward range of the game. This suggests that assuming last-iterate convergence is not unreasonable for our setting.

## C.8 STRATEGY PURIFICATION

As mentioned in the body, we partially *purify* our strategy before playing. When *Maxmargin* is used as the subgame solving algorithm (*i.e.*, when the margins are all nonnegative), we allow mixing between $k = 3$ actions; when *Resolve* is used, we deterministically play the top action. Moreover, we only allow mixing among actions other than the highest-probability action if they have appeared continuously in the support of $x^t$ for every iteration $t > T_{1/2}$, where $T_{1/2}$ is chosen to be the iteration number when half the time budget elapsed.[19] We call such actions "stable". These restrictions reduce the chance that transient fluctuations in the strategy of the player, which occur commonly during game solving especially with an algorithm like PCFR+, would affect the final action that is played. Any probability mass that was assigned to actions that are excluded in the above manner is shifted to the action with highest probability.

## D HARDWARE

*Obscuro*, for its human matches, ran on a single desktop machine with a 6-core Intel i5 CPU. Ablations and further matches were run on an AMD EPYC 64-core server machine using 10 cores (5 per side). We now report statistics about the computational performance of *Obscuro*. These statistics were collected over the course of a 1,000-game sample, at a time control of 5 seconds per move.[20]

- Average game length: 116.6 plies (58.3 full moves)

---

[19]It will almost always be the case that $T_{1/2} < T/2$. This is because, as the game tree grows larger, PCFR+ iterations, whose time complexity scales with the size of the game tree, get slower.

[20]For this and all other AI-vs-AI matches in this paper, the stated time control, usually 5 seconds per move, is the time limit allocated to the main search loop, and does *not* include the time it takes to enumerate the set of all legal positions.

- Average search depth: 10.7 plies

- Average search tree size: 1,070,552 nodes, 14,404 infosets

- Average search tree size carried over from previous search: 181,421 nodes, 3,162 infosets

- Average number of possible positions: 17,264

## E    OBSERVATIONS ABOUT FoW CHESS

### E.1    SIZE OF INFOSETS AND COMMON-KNOWLEDGE SETS

Here we elaborate on the discussions about common-knowledge sets and infosets, alluded to in the introduction.

Consider the family of positions in which both sides have spent the first eight moves playing **1. a4 a5 2. b4 b5 ... 8. h4 h5**, and subsequently shuffle all their remaining pieces around their first three ranks. An example of such a position is in Fig. 5. Each player must have one bishop on a light square (12 ways), one bishop on a dark square (12 ways), one queen, one king, two knights, and two rooks ($22 \cdot 21 \cdot 20 \cdot 19 \cdot 18 \cdot 17/2^2$ ways). When multiplied, this gives a total of approximately $M = 2 \times 10^9$ ways. This is a lower bound on the maximum size of an infoset. For common-knowledge sets, *both* players can arrange their pieces arbitrarily along the first three ranks, yielding approximately $M^2 \approx 4 \times 10^{18}$ different arrangements, which provides a lower bound on the maximum size of a common-knowledge set.[21]

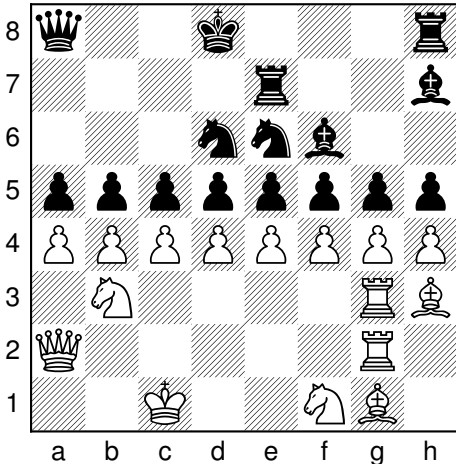

Figure 5: FoW chess position illustrating the existence of large infosets and common-knowledge sets. A full explanation is given in the text.

Although infosets *can* get this large, they almost never *do* in practical games, because both sides are making effort to obtain information.

We now elaborate on Fig. 1. In particular, we will show that the two positions in that figure are in the same common-knowledge set. Consider the sequence of positions in Fig. 6, read in order from top-left to bottom-right. The positions marked A and B are the same as those in in Fig. 1. Each position is connected to the next one by an infoset of one of the players: the first pair by a White infoset, the second pair by a Black infoset, and so on. A computer search showed that the depicted path, which

---

[21]These common-knowledge sets are measured with respect to *states*, not *histories*. Measuring common-knowledge sets with *histories* would result in a significantly larger number, because the order of the moves would matter.

has length 9, is the shortest path between these two positions.[22] Hence, if the true position is A, then the statement $Y$ = "The true position is not B" is 8th-order knowledge for both players. That is, it is true that

$$\underbrace{\text{everyone knows everyone knows ... everyone knows}}_{\text{8 repetitions}} Y$$

yet the same statement would be false if there were 9 repetitions, so $Y$ is not common knowledge.

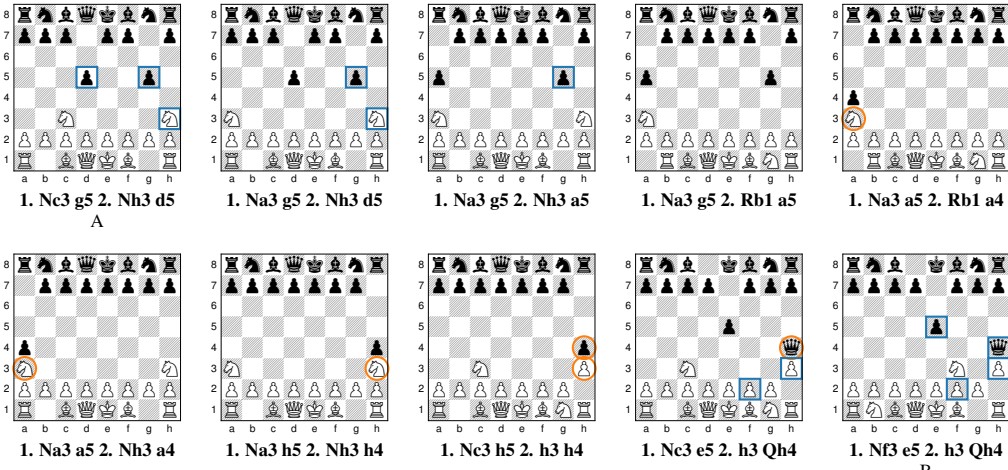

Figure 6: Sequence of positions illustrating the connectivity between the two positions in Fig. 1. Circles mark squares that the opponent knows are occupied by *some* piece, but not by *which* piece. A full explanation is given in the text.

### E.2    MIXED STRATEGIES

Playing a mixed strategy is a fundamental part of strong play in almost any imperfect information game, and it is particularly important in games like FoW chess where there is no private information assigned by chance, such as private cards in poker. Indeed, in small poker endgames, deterministic strategies exist for playing near-optimally (Farina & Sandholm, 2022). However, in FoW chess, if a player plays a pure strategy that the opponent knows, the opponent would essentially be playing regular chess, because the opponent can predict with full certainty what the player would play. This is a significant disadvantage that will result in a rapid loss against any competent opponent.

Consider, for example, the position in Fig. 7(A). White can win almost a full pawn (in expectation) by mixing between the moves **2. Qa4** with low probability and **2. Nc3** with high probability. No move for Black simultaneously defends the threats against both the king and the pawn. (**2... c6** may look like it does, but after **3. cxd5**, Black cannot recapture the pawn without risking hanging a king or queen.)[23]

This necessity of playing a mixed strategy explains why we do not adopt full purification of our strategy and instead opt to allow mixing.

### E.3    FIRST-MOVER ADVANTAGE

We evaluated the first-mover advantage in FoW chess by running 10,000 games with *Obscuro* playing against itself at a time control of 5 seconds per move. Of these games, White scored 57.5% (+4935 =1623 -3442). This is, with statistical significance ($z > 5$), larger than the empirical first-move advantage in regular chess, which is about 55% (Chessgames.com, 2024). We believe that the

---

[22]A similar computer search shows that this is nearly the longest possible shortest path between any pair of nodes after two moves from each side: there is a shortest path of length 10, but no shortest paths longer than that.

[23]*Obscuro* prefers to also include **3. Nf3** and **3. e3** in its mixed strategy to dissuade **2... d4**.

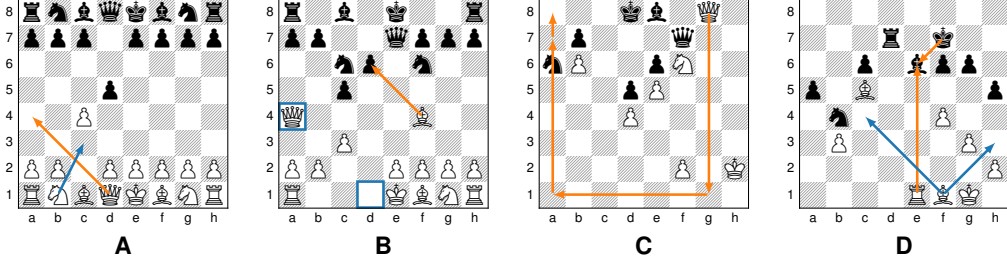

Figure 7: FoW chess positions from actual gameplay illustrating common themes. **(A)** Opening position after the common trap **1. c4 d5?!** **(B)** An early-game bluff. White bluffs that its attacking bishop is defended by the queen on **d1**. **(C)** A highly-risky queen maneuver from a losing position. **(D)** An endgame position in which the disadvantaged side sacrifices material for a chance at the opposing king. Details can be found in the text.

| | White | | Black |
|---:|---:|---:|---:|
| **d4** | 66.4% | **Nc6** | 32.5% |
| **c4** | 29.6% | **c6** | 25.1% |
| **e4** | 1.9% | **e6** | 20.7% |
| **Nc3** | 1.4% | **Nf6** | 15.9% |
| **c3** | 0.4% | **c5** | 4.8% |
| **Nf3** | 0.2% | **d5** | 0.9% |

Table 1: Distribution of first moves played by *Obscuro* as both White and Black, over a 10,000-game sample. Percentages may not add up to 100% due to rounding.

fundamental reason for this discrepancy is the weakness of the **a4-e8** diagonal, as already exhibited in Fig. 7(A), discussed above. This risk presents Black from developing in a natural manner against **1. c4** or **1. d4**, allowing White a healthy opening lead.

Indeed, our 10,000-game sample included 10 games with length 12 ply (6 moves from each player) or fewer; all 10 of these games ended with either Black failing to cover **Qa4+** or White failing to cover **Qa5+**:

- **1. c4 d5 2. Qa4+ d4** 1-0

- **1. c4 c6 2. d4 d5 3. cxd5 Qa5+ 4. Qa4** 0-1

- **1. c4 Nc6 2. d4 d5 3. Qa4 dxc4 4. d5 Nb8** 1-0

- **1. d4 c6 2. c4 d5 3. cxd5 Bf5 4. Qa4 cxd5** 1-0  (*This play-through occurred three times.*)

- **1. d4 c6 2. c3 e6 3. e4 d5 4. e5 c5 5. Qa4+ cxd4** 1-0

- **1. c4 e6 2. d4 c5 3. d5 Qa5+ 4. Nd2 Nf6 5. e4 Nxe4 6. Nxe4** 0-1

- **1. d4 c6 2. Nc3 d5 3. Qd3 Nf6 4. e4 dxe4 5. Nxe4 Qa5+ 6. Nxf6+** 0-1

- **1. c4 d5 2. Qa4+ c6 3. cxd5 Nf6 4. dxc6 Nxc6 5. Nf3 e5 6. Nxe5 Nxe5** 1-0

These games may seem like they contain major mistakes, but that is not so. It is rather likely that *most or all of these play-throughs are part of optimal play*: after all, bluffs must sometimes get called!

In Table 1 we give *Obscuro*'s mixed strategy on the first move for both White and Black, over the 10,000-game sample. The above observation about the **a4-e8** diagonal has a large effect on opening choices. We believe that this explains why White strongly prefers opening with **d4** and **c4** rather than **e4** which is equally favored in regular chess, and why Black almost never opens with **d5** and instead prefers to immediately close the dangerous diagonal by moving something to **c6**.

### E.4 BLUFFS

*Obscuro* bluffs. An example bluff is in Fig. 7(B), which is from the aforementioned 10,000-game sample. White knows that **d6** is defended (in fact, White knows the exact position). Black does not know the location of the white queen (for example, it could be on **d1** instead). This allows White to play **Bxd6**, exploiting the fact that Black cannot recapture without risking losing the queen.

### E.5 PROBABILISTIC TACTICS AND RISK-TAKING

The existence of hidden information in FoW chess allows tactics that would not work in regular chess. An example of this phenomenon as early as move 2 has already been described above, where mixing allows White to win a pawn after **1. c4 d5**. We now give additional examples.

Fig. 7(C) depicts a position encountered during our 20-game match against the top-rated human. *Obscuro* (White) was in a losing position, down a minor piece. It decided to play the highly risky queen maneuver **Qg8-g1-a1-a7-a8**, leaving its own king exposed in order to attempt to hunt the opposing king. This risky tactic worked: the game played out **68. Qg1 Qe7 69. Qa1 Nb8 70. Qa7 Nd7 71. Qa8+ Nxb6** 1-0.[24] This sequence of moves heavily exploits the opponent's imperfect information: if Black knew that White was attempting this attack, Black could easily either defend the attack or launch a counterattack on the completely undefended white king.

For another example, consider the position in Fig. 7(D), again from the aforementioned 10,000-game sample, and suppose for the sake of the example that White has perfect information. White faces a slight material disadvantage in an endgame. However, *Obscuro* as White finds the tactical blow **1. Rxe6! Kxe6** upon which mixing evenly between **2. Bc4+** and **2. Bh3+** wins on the spot with 50% probability.

### E.6 EXPLOITATIVE VS. EQUILIBRIUM PLAY

The position in Fig. 7(D) is also an example of the difference between *exploitative* play and *equilibrium* play in FoW chess. The above tactic has expected value at least 50% against any player, because it wins on the spot with probability at least 50%. It is likely the best move if playing against a perfect opponent. However, against a substantially weaker player, it may be far from the best move: against a weak player, one can argue that the endgame is probably a win even with the slight material disadvantage, whereas the tactic will lead to a significant disadvantage (down three points of material) if it fails to win. Therefore, if one knew the strength of one's opponent, one may opt to not go for this tactic and instead attempt to win the endgame in a "safer" manner. Another example of this phenomenon was also seen above. *Obscuro*, with small probability, can lose in two moves (**1. c4 d5 2. Qa4+ d4**). Any player, no matter how weak, can therefore beat *Obscuro* with positive probability as White by simply playing the above move sequence. However, against opponents below a certain level, playing the above moves as Black may be considered a needless risk.

*Obscuro* does not know or attempt to model the opponent. It will simply play what it believes to be a near-equilibrium strategy. Therefore, it may not do as well against weak players as an agent designed specifically to exploit weak players. This design choice was intentional, and follows other efforts in superhuman game-playing AI such as those mentioned in the introduction, most of which attempted to find and play equilibria rather than to exploit a particular opponent.

### E.7 VOLATILITY

FoW chess is a highly volatile, highly stochastic game. Indeed, the previous two observations regarding risk taking and exploitative play are evidence of this. Most games, including a majority of our 20 games against the world #1 player, are ultimately decided by one side outright "blundering material" because of lack of knowledge of the opponent's position. We emphasize, however, that this is not a sign of poor quality of play; rather, we believe that strong play in FoW chess involves calculated risk-taking that, with nontrivial probability, leads to such "blunders". More skilled play-

---

[24]The immediate **70. Qa8** would have worked in this position as well, but it was not played, likely because it would have risked losing the queen in case the king were on **b8**.

ers are better at taking calculated risks while restricting the probability of losing material, and at forcing their opponents into more risky situations.

### E.8 KING VS KING

To make some of the above discussions about mixing, volatility, and equilibrium play more concrete, we include here a partial analysis of the king-vs-king endgame, assuming the starting position of the kings is common knowledge. While this endgame is an immediate draw in the rules of regular chess (because a lone king cannot checkmate), FoW chess allows such endgames to play out, and not all such endgames are immediately drawn; in fact, the analysis turns out rather intricate already. In the below discussion, 0 is a draw, $+1$ is a certain win for White, and $-1$ is a certain win for Black.

**Claim 1.** *Suppose that there are two legal moves for the black king that are 1) guaranteed to be safe (i.e., guaranteed to not immediately lose), and 2) adjacent to each other (orthogonally or diagonally). Then Black secures at least a draw.*

*Proof.* Black randomly moves to one of them on their first move, and shuffles between them forever thereafter. The white king cannot approach without being captured half of the time. $\square$

Thus, it remains only to discuss the case where one king is on the edge of the board. Assume, without loss of generality, that this is the black king, and that it is on the 8th rank.

**Claim 2.** *If the white king prevents the black king from immediately moving off the back rank (e.g., **a6** and **a8**), the equilibrium value is* strictly *positive, regardless of which side is to move.*

*Proof.* We will show that Black has no strategy that achieves expected value 0. Consider two cases.

*Case 1.* Black's strategy involves attempting to move off the back rank with positive probability on some move $t$ (but not earlier). Then consider the following strategy for White. Let $x\mathbf{7}$ (for $x \in \{\mathbf{a}, \mathbf{b}, ..., \mathbf{h}\}$) be the square on the 7th rank with maximal probability $p > 0$ for the black king after $t$ moves. White places its king on square $x\mathbf{6}$ before Black's $t$th move. With probability $p$, White wins immediately. Otherwise, White runs away downwards, executing the strategy from Claim 1, forcing a draw.

*Case 2.* Black's strategy is to always stay on the back rank. Then consider the following strategy for White. Let $x\mathbf{8}$ be the square on the back rank with *minimal* probability $q \leq 1/4$ for the black king, at the time when White makes its 8th move. White places its king on $x\mathbf{7}$ on its 8th move, then moves left and right on the 7th rank until it wins. We claim that White has expected value at least $1 - 2q = 1/2$ with this strategy. To see this, note that, since Black always stays on the back rank, the *parity* of its rank alternates between moves; therefore, if the black king is *not* on $x\mathbf{8}$, then White will not lose on its 8th move. Further, also by a parity argument, White will eventually chase down the black king and win the game. $\square$

If the black king is on the edge of the board, it is always the case that either White can force the kings to be two squares apart with common knowledge (Claim 2) or Black has a safe pair of adjacent moves (Claim 1), so this completes the analysis.

We complete this section by pointing out an interesting special case: If the black king starts in the corner (**a8**), the white king starts on either **b6, c7**, or **c6**, and it is White to move, then White can secure value strictly larger than $1/2$: randomize between **Kb6, Kc7, Ka6**, or **Kc8** (whichever are legal moves) on the first move. This wins with probability $1/2$ immediately, and otherwise immediately forces the kings to be two squares apart (Claim 2).

```
 1: maintain
 2:     current game tree Γ (initially containing only the root ∅)
 3:     current strategy profile (x, y)
 4:     current expected value v*
 5:     current set of possible positions P (initially containing only the root ∅)
 6: procedure MOVE(observation sequence o)
 7:     CONSTRUCTSUBGAME(o)
 8:     in parallel
 9:         RUNSOLVERTHREAD()
10:         RUNEXPANDERTHREAD() ▷ or multiple parallel copies of expansion thread
11:     I ← our current infoset
```
12: $\quad a^* \leftarrow \operatorname{argmax}_{a \in A(I)} \pi(a|I)$ ▷ ties broken arbitrarily
13: $\quad$ **if** $p_{\max} = 0$ **then** ▷ If using Resolve, just play $a^*$, *i.e.*, purify completely.
14: $\quad\quad S \leftarrow$ set of stable actions $\cup \{a^*\}$ ▷ "Stable" is defined in the text.
15: $\quad\quad$ **if** $|S| >$ MAXSUPPORT **then** ▷ The parameter MAXSUPPORT is set to 3.
16: $\quad\quad\quad$ remove all but the top MAXSUPPORT most likely actions in $S$
17: $\quad\quad \pi_{\text{play}}(\cdot|I) \leftarrow \pi(\cdot|I)$
18: $\quad\quad$ **for** action $a \in A(h) \setminus S$ **do** ▷ Shift all mass of such actions onto $a^*$.
19: $\quad\quad\quad \pi_{\text{play}}(a^*|I) \leftarrow \pi_{\text{play}}(a^*|I) + \pi_{\text{play}}(a|I)$
20: $\quad\quad\quad \pi_{\text{play}}(a|I) \leftarrow 0$
21: $\quad$ sample $a^* \sim \pi(\cdot|I)$
22: $\quad$ play action $a^*$

Figure 8: Pseudocode, Part 1.

```
 1: procedure CONSTRUCTSUBGAME(observation sequence o)
 2:     ▷ The set of possible positions is updated on every move by simply enumerating
 3:     ▷     all possibilities
 4:     P ←  all positions consistent with o
 5:     I ←  set of nodes in Γ consistent with o
 6:     ▷ Construct KLUSS subgame:
```
7: $\quad$ **for** each opponent infoset $J \subseteq \overline{I^2}$ **do**
8: $\quad\quad$ set alternate value $v^{\text{alt}}(J) \leftarrow u(x, y|J) - \hat{g}(J)$
9: $\quad$ **while** $|I| < \min\{|P|, \text{MININFOSETSIZE}\}$ **do** ▷ Add more states to $I$ if there are not enough.
10: $\quad\quad$ ▷ The parameter MININFOSETSIZE is set to 256.
11: $\quad\quad$ get random state $s \in P \setminus I$
12: $\quad\quad$ ▷ Assume ▼ has perfect information at newly-sampled states:
13: $\quad\quad$ add $s$ to ▲-infoset $I$ and ▼-infoset $J = \{s\}$
14: $\quad\quad$ set alternate value $v^{\text{alt}}(J) \leftarrow \min\{\tilde{v}_▲(s), v^*\}$
15: $\quad$ ▷ Fix prior probabilities:
16: $\quad \mathcal{J}_0 \leftarrow \{J : J \subseteq \overline{I^2}\}$
17: $\quad$ **for** each opponent infoset $J \subseteq \overline{I^2}$ **do** set prior probability $\alpha(J) \leftarrow \frac{1}{2}\left(\frac{y(J)}{\sum_{J'} y(J')} + \frac{1}{m}\right)$
18: $\quad$ create new root node $\varnothing$ where ▼ selects infoset $J \in \mathcal{J}_0$, reaching node $h_J$
19: $\quad$ ▷ $\pi_{-i}(h)$ is the probability that all other players play all actions on the path to $h$ in Γ.
20: $\quad$ **for** each $J \in \mathcal{J}_0$ **do**
21: $\quad\quad$ make $h_J$ a chance node where $h \in J$ is selected w.p. $\propto \pi_{-▼}(h)$
22: $\quad\quad$ make new regret minimizer $R_J$ with strategy space $[0, 1]$ using PRM+
23: $\quad\quad$ ▷ Regret minimizer $R_J$ controls the probability with which ▼ enters at $J$ in Resolve.
24: $\quad \Gamma \leftarrow$ game tree with root $\varnothing$
25: $\quad$ delete all game tree nodes not reachable from $\varnothing$

Figure 9: Pseudocode, Part 2.

```
 1: procedure RUNSOLVERTHREAD
 2:     while time permits do ▷ Run for longer than expander threads.
 3:         RUNCFRITERATION(▲)
 4:         RUNCFRITERATION(▼)
 5:         ▷ Special case, must be done separately: RM+ updates for the Resolve subgame:
 6:         for each J ∈ 𝒥₀ do perform regret minimizer update at R_J with utility u_▼^cf(J)
 7:         ▷ Transition between Resolve and Maxmargin smoothly,
 8:         ▷     based on whether Resolve chooses to enter at any infoset J:
 9:         ▷ π_▼^resolve(J) is the probability that Resolve (R_J) enters at J.
10:         ▷ π_▼^maxmargin(J) is the probability Maxmargin picks J.
11:         p_max ← max_{J∈𝒥₀} π_▼^resolve(J)
12:         for each J ∈ 𝒥₀ do π_▼(J) ← p_max · α(J) · π_▼^resolve(J) + (1 − p_max) · π_▼^maxmargin(J)
13:         ▷ Note: it is possible for ∑_{J∈𝒥₀} π_▼(J) ≠ 1 if Resolve is being used!

14:
15: procedure RUNCFRITERATION(exploring player i)
16:     MAKEUTILITIES(i, ∅) ▷ MAKEUTILITIES will mark some infosets VISITED.
17:     ▷ u_▼^cf(J) is ▼'s CFV for picking J at the root.
18:     if i = ▼ then
19:         for each J ∈ 𝒥₀ do u_▼^cf(J) ← u_▼^cf(J) + v^alt(J)
20:     for each VISITED infoset I, in bottom-up order do
21:         ▷ π_i is player i's strategy. σ(I) is the parent sequence of infoset I.
22:         ▷ CFR value backpropagation:
23:         u_i^cf(σ(I)) ← u_i^cf(σ(I)) + ∑_{a∈A(I)} π_i(a|I)u_i^cf(I, ·)
24:         perform regret minimizer update at I using counterfactual values u_i^cf(I, ·)
25:         mark I not VISITED
26:         u_i^cf(I, ·) ← 0 ▷ reset
```

Figure 10: Pseudocode, Part 3.

```
 1: procedure MAKEUTILITIES(exploring player i, node h)
 2:     mark h as not NEW
 3:     if h is not EXPANDED or h is terminal then
 4:         (I, a) ← σ_i(h)
 5:         mark I as VISITED
 6:         ▷ ṽ_i(h) is the Stockfish evaluation or terminal node value of h from i's perspective.
 7:         ▷ u_i^cf(I, a) stores the CFV at sequence (I, a). Initialized to 0.
 8:         u_i^cf(I, a) ← u_i^cf(I, a) + π_{-i}(h)ṽ_i(h)
 9:     else
10:         ▷ No need to explore nodes to which the opponent does not play.
11:         ▷ No locks needed: all EXPANDED nodes are safe to access.
12:         for each legal action a at h do
13:             if i plays at h or π_{-i}(ha) > 0 then MAKEUTILITIES(i, ha)
```

Figure 11: Pseudocode, Part 4.

```
1: procedure RUNEXPANDERTHREAD
2:     while time permits do
3:         DOEXPANSIONSTEP(▲)
4:         DOEXPANSIONSTEP(▼)
5:
6: procedure DOEXPANSIONSTEP(exploring player i)
7:     h ← root node of current subgame Γ
8:     while h is EXPANDED do ▷ Find leaf to expand.
9:         ▷ Terminal nodes cannot be expanded.
10:        ▷ Also, we should expand nodes that CFR has not yet iterated on.
11:        if h is terminal or h is NEW then return
12:        ▷ Select action:
13:        ▷ π̃_i is the expansion strategy of player i as defined in the text.
14:        for action a ∈ A(h) do
15:            if h belongs to i then π̃(a|h) ← π̃_i(a|h)
16:            else π̃(a|h) ← π_{-i}(a|h)
17:        ▷ If h = ∅ and ▼ is using Resolve, π̃(·|h) may not be a distribution
18:        sample a ∈ A(h) w.p. ∝ π̃(·|h)
19:        h ← ha
20:    ▷ Expand h:
21:    j ← active player at ha
22:    add all children of h to Γ
23:    let I be the infoset that h should be in
24:    if I is not created then
25:        create I
26:        initialize current strategy as π_j(a*|I) = 1 where a* := argmax_{a∈A(h)} ṽ_j(ha)
27:    add h to I
28:    mark h as EXPANDED
```

Figure 12: Pseudocode, Part 5.

