# OpenReview forum: "General search techniques without common knowledge for imperfect-information games, and application to superhuman Fog of War chess"
_ICLR.cc/2026/Conference — ICLR 2026 Poster_

### Official Review · Reviewer_uPu3 · 2025-10-31

**Soundness:** 4
**Presentation:** 3
**Contribution:** 3
**Rating:** 8
**Confidence:** 4

**Summary:**

The paper presents a game-playing agent, named Obscuro. Obscuro is the first artificial agent that has achieved superhuman performance in Fog-of-war Chess, an imperfect information variant of Chess. Obscuro is a fully search-based agent that builds on the previous state-of-the-art agent in FoW Chess. The main algorithm behind Obscuro, Knowledge-limited Unfrozen Subgame Solving (KLUSS), is a generalization of KLSS. The paper reports good results against the previous state-of-the-art agent, human players of various skill levels, as well as the number one-ranked human player. Finally, the proposed improvements and their importance to the final performance of Obscuro are validated in a series of ablation experiments.

The paper is clearly written overall, with a well-motivated introduction, detailed algorithmic description, and a comprehensive experimental section that includes strong evaluations against both prior state-of-the-art methods and human players. The ablation studies effectively validate the importance of the proposed improvements. However, the clarity is weakened by missing or inconsistent details, such as incorrectly labeled figures, missing descriptions of the subgame solving algorithm in the main text, and ambiguities in the ablation setup. Some claims, like FoW Chess being the largest imperfect-information game where search has been applied, seem unsupported.

**Strengths:**

**Originality.**
The experiments convincingly support the claim that Obscuro is the first AI agent to achieve superhuman performance in FoW Chess.

**Clarity**
The whole paper is nicely and concisely written. The introduction motivates the need for scalable search in imperfect information games and the reasons it’s substantially harder than in perfect information games.
The agent’s description is clearly and thoroughly written, providing detailed explanations of each component.

**Soundness.**
The experimental section is comprehensive, including detailed evaluations against both previous state-of-the-art methods and human players of different skill levels, up to the top-rated player on chess.com, providing strong empirical evidence for the quality of the proposed method.
The ablation studies consistently demonstrate performance improvements of the proposed modifications, confirming that the improvements are both well-justified and impactful.

**Weaknesses:**

**Clarity.**
Several parts of the paper suffer from unclear or inconsistent presentation:
* Lines 216–220 introduce a connectivity graph with nodes labeled by their distance from the circled node and one node marked with an asterisk, yet no such labeling appears in the corresponding figure, making it difficult to follow the explanation.
* The description of the main algorithm omits details about the subgame solving algorithm, describing it only in the appendix.
* It is not clear which opponent was used in the ablation experiments and what the “above four improvements” in the fourth ablation (GT-CFR Only) refer to
* It is also not clear which components were and were not used in some of the ablations

**Soundness.**
The claims that FoW Chess is the largest (measured by the amount of hidden information) turn-based game and the largest imperfect information game where search has been successfully applied are not based on any concrete numbers comparing the game to other such games.
The first sentence in the conclusion is misleading, as Obscuro requires a strong value function to reach such a strong play (as shown by one of the ablation experiments).

**Questions:**

* Is the set $P$ from Section 3 an infoset of the acting player?
* Are the ablations run against the previous state-of-the-art algorithm for FoW Chess?
* Do you have any intuition why two-sided GT-CFR scored so low as compared to one-sided GT-CFR in the ablations?

Minor comments:
* Line 53: A missing citation to DeepStack
* Lines 147-150: The problem of deciding whether two histories belong to the same common-knowledge closure has been studied by Solinas et al. [1]

[1] Solinas, Christopher, et al. "History filtering in imperfect information games: algorithms and complexity." Advances in Neural Information Processing Systems 36 (2023): 43634-43645.

---

> ### Author Response · Authors · 2025-11-22
>
> Thank you for your review.
>
> > Lines 216–220 introduce a connectivity graph with nodes labeled by their distance from the circled node and one node marked with an asterisk, yet no such labeling appears in the corresponding figure, making it difficult to follow the explanation.
>
> The nodes in the first layer (the one that includes the circled node) are supposed to be labeled (from left to right) $0, 1, 2, 3, *$. Apologies for the error; it is fixed in the revision.
>
>
>
> > Soundness. The claims that FoW Chess is the largest (measured by the amount of hidden information) turn-based game and the largest imperfect information game where search has been successfully applied are not based on any concrete numbers comparing the game to other such games.
>
> Imperfect-information search techniques for zero-sum  games, to our knowledge, have been applied only in poker and FoW chess.  As we discuss in the body, *public states* in two-player Texas hold'em poker have size $10^3$, whereas single information sets in FoW chess easily exceed $10^6$ positions often in practice, and can be larger than $10^{9}$ in theory (Appendix E.1 in the revision).
>
> We have revised to clarify the "zero-sum" caveat to these claims.
>
> > The first sentence in the conclusion is misleading, as Obscuro requires a strong value function to reach such a strong play (as shown by one of the ablation experiments).
>
> We have revised to clarify this.
>
> > It is not clear which opponent was used in the ablation experiments and what the “above four improvements” in the fourth ablation (GT-CFR Only) refer to.
> It is also not clear which components were and were not used in some of the ablations
>
> We have revised Section 4.1 in the hope of clarifying these things.
>
> "Above four improvements" was an error of ordering; the non-uniform resolve distribution was also off for this experiment. That ablation serves to isolate the (significant!) effect of switching from a procedure based on LP and iterative deepening (as used by ZS21) to *two-sided* GT-CFR.
>
> > Are the ablations run against the previous state-of-the-art algorithm for FoW Chess?
>
> Unless otherwise stated, all ablations were *Obscuro* playing against a version of *Obscuro* with a single technique turned off. This is now clarified in the revised version.
>
> > Do you have any intuition why two-sided GT-CFR scored so low as compared to one-sided GT-CFR in the ablations?
>
> Two-sided GT-CFR scored 46.7% against one-sided GT-CFR (this is Ablation #3). This is not particularly low. Perhaps the above clarification regarding what is now Ablation #5 answers your question?
>
> > Is the set $P$ from Section 3 an infoset of the acting player?
>
> Almost, but not quite: an information set should contain histories, while $P$ contains positions, which are not the same.
>
>
> > Line 53: A missing citation to DeepStack
> > Lines 147-150: The problem of deciding whether two histories belong to the same common-knowledge closure has been studied by Solinas et al. [1]
>
>
> We have added the suggested citations.

---

### Official Review · Reviewer_abjR · 2025-10-31

**Soundness:** 4
**Presentation:** 3
**Contribution:** 4
**Rating:** 8
**Confidence:** 3

**Summary:**

The paper presents Obscuro, the first AI to reach a superhuman level in the chess variant Fog of War chess (FoW), and reaches a new state-of-the-art for this game type.
The main innovation lies in its new search adaptations.
In fact, the agent is entirely based on real-time search, rather than training a new neural network policy or value function; it relies on classical Stockfish 14 for standard chess for their node evaluation.
Related to search, they introduce knowledge-limited unfrozen subgame solving (KLUSS) search as an improvement over the KLSS algorithm. Next, they utilize growing-tree counterfactual regret minimization (GT-CFR), modifying it to one-sided GT-CFR. Here, they use predictive CFR+ (PCFR+) for equilibrium finding. To expand the game tree, they utilize the polynomial upper confidence bounds for trees (PUCT) algorithm.
The authors also provide game footage of their agents, along with an elaborate appendix that explains the search techniques in detail.

**Strengths:**

+ The engine runs on regular consumer hardware (the details are provided in Appendix B.5.).
(Preferably, refer to Appendix B.5 directly in the main paper.)

+ The presented agent Obscuro wins both against the previously best FoW engine and the strongest human player on chess.com by a significant margin (+302 +/- 29 Elo over ZS21, +241 +/- 274 Elo over top human).

+ The paper presents numerous ablation studies that highlight the benefits of their individual modifications to existing algorithms.

+ The paper is well formulated and guides the reader throughout the content.

+ It includes a detailed appendix with pseudocode that provides a detailed explanation of the search techniques.

Relevancy:
The paper appears to be highly relevant for a major audience.

Overall Assessment
Overall, I believe it is a great paper that pushes the state-of-the-art in the field of imperfect information games to a new level.

**Weaknesses:**

- "required no large-scale computation to learn a value function or blueprint strategy"
-> This is true, but the evaluation function was taken from Stockfish 14, which underwent a long period of development. The authors also show in their ablation studies that the evaluation function plays a significant role in their playing strength.
Maybe clarify this statement a bit.

- The authors do not mention whether they will make their source code and/or engine publicly available.

- It was not directly clear to me at first sight in which perspective the published games were, i.e. if Obscuro was White or Black. -> Please add a comment in the paper that the games are always in the view of Obscuro.

- The sample size (20 games  (+16 =0 -4), +241 +/- 274 Elo) against the top human was not large enough to 100% claim superiority over the top human. You may also provide the relative Elo superiority with error bounds in the main paper. The top human could also try to find fundamental weaknesses in its play over time. This seems, however, not likely as it attempts to find a near-equilibrium strategy.
I also consider 20 games to be a good sample size, given it's hard to find top players playing an engine.

- The paper would benefit from an additional evaluation environment despite FoW, to highlight the broader applicability of their search agent. This alternative environment could be poker or Stratego for example.

Minor things:
"Liu et al. (Liu et al., 2023) introduced a safe variant of KLSS"
-> Here, Liu et al. is mentioned twice. You can avoid this by using \citet{}

**Questions:**

"4. Two-sided GT-CFR only, against ZS21. In this ablation, we turned off all the above im-
provements 1, 2, 3, and 4, and matched the resulting agent against that of ZS21. This
serves to isolate the effect of using GT-CFR compared to using the LP-based equilibrium
computation and iterative deepening node expansion as in ZS21.
In a 1,000-game match, the GT-CFR version scored 72.6% (+711 =30 -259)."

-> This is a bit unexpected to score so highly here, when turning off all 3 previous improvements, right?
I likely misunderstood what was done here.

---

> ### Author Response · Authors · 2025-11-22
>
> Thank you for your review.
>
> > "required no large-scale computation to learn a value function or blueprint strategy" -> This is true, but the evaluation function was taken from Stockfish 14, which underwent a long period of development. The authors also show in their ablation studies that the evaluation function plays a significant role in their playing strength. Maybe clarify this statement a bit.
>
> We have revised to clarify this.
>
> > The authors do not mention whether they will make their source code and/or engine publicly available.
>
> We do not intend to publish our source code or engine. Other major prior superhuman game-playing milestones (chess, Go, two-player limit Texas hold'em poker, two-player no-limit Texas hold'em, multi-player no-limit Texas hold'em, etc.) also did not do so, for varying reasons including the danger of allowing human players easy access to a method of cheating. We include, in the appendix, detailed pseudocode that should permit replication and extension.
>
> > It was not directly clear to me at first sight in which perspective the published games were, i.e. if Obscuro was White or Black. -> Please add a comment in the paper that the games are always in the view of Obscuro.
>
> We have revised the paper to clarify this.
>
> > The sample size (20 games (+16 =0 -4), +241 +/- 274 Elo) against the top human was not large enough to 100% claim superiority over the top human. You may also provide the relative Elo superiority with error bounds in the main paper. The top human could also try to find fundamental weaknesses in its play over time. This seems, however, not likely as it attempts to find a near-equilibrium strategy. I also consider 20 games to be a good sample size, given it's hard to find top players playing an engine.
>
> We gave the human ample time to poke the bot for weaknesses: 20 games is a decent amount of data, and the games were played over the course of two days, 10 games per day---so our human opponent can (and did) analyze the first set of games games overnight.
>
> We're not sure where the "+/- 274 Elo" error bar is coming from; assuming that this is a 95% confidence interval, it would suggest that our result is not statistically significant at the 95% level, but an exact two-sided binomial test gives a p-value of $\frac{2}{2^{20}} \sum_{k=16}^{20} \binom{20}{k} \approx 0.0118$ (apologies, the previous version incorrectly rounded this to 0.011; it is now fixed.) Of course, no statistical test is ever "100%" perfect, but we believe that this is sufficient evidence of superiority.
>
> > "4. Two-sided GT-CFR only, against ZS21. In this ablation, we turned off all the above im- provements 1, 2, 3, and 4, and matched the resulting agent against that of ZS21. This serves to isolate the effect of using GT-CFR compared to using the LP-based equilibrium computation and iterative deepening node expansion as in ZS21. In a 1,000-game match, the GT-CFR version scored 72.6% (+711 =30 -259)."
>
> "Above four improvements" was an error of ordering; the non-uniform resolve distribution was also off for this experiment. This has been fixed in the revised version.
>
> >  This is a bit unexpected to score so highly here, when turning off all 3 previous improvements, right? I likely misunderstood what was done here.
>
> This experiment shows that GT-CFR, even the two-sided version, far outperforms an exact LP + iterative deepening approach to node expansion, and that using GT-CFR instead of linear programming was a significant contributing factor to the improved performance of *Obscuro* over the prior SOTA.

---

> > ### Comment · Reviewer_abjR · 2025-11-26
> > **Thank you for fixes and elaborations**
> >
> > Thank you for your elaborations and fixes.
> > The Elo error was computed using a 95% confidence interval:
> > https://3dkingdoms.com/chess/elo.htm
> > The same formula is also used in cutechess:
> > https://github.com/cutechess/cutechess/blob/master/projects/lib/src/elo.cpp#L44
> >
> > I keep my score at "8: accept, good paper (poster)".

---

### Official Review · Reviewer_3cmb · 2025-11-01

**Soundness:** 2
**Presentation:** 1
**Contribution:** 3
**Rating:** 6
**Confidence:** 3

**Summary:**

This paper introduces Obscuro, an AI system to achieve superhuman performance in Fog of War (FoW) Chess, a long-standing benchmark in imperfect-information game solving. The key technical contribution is a suite of scalable real-time search techniques—most notably Knowledge-Limited Unfrozen Subgame Solving (KLUSS) and one-sided GT-CFR—that circumvent the need to compute or enumerate common-knowledge sets, which are intractable in FoW Chess (up to ~10¹⁸ states). The authors demonstrate Obscuro’s superiority through extensive experiments: it wins 85.1% of 1,000 games against the prior SOTA (ZS21) and 80% of 20 games against the world’s top-ranked human player (rating 2318 on chess.com).

**Strengths:**

Significant Technical Advance: The paper directly addresses a fundamental scalability bottleneck in imperfect-information game solving—the reliance on common knowledge—and proposes practical, theoretically motivated approximations (KLUSS) that enable real-time search in previously intractable settings.

Strong Empirical Validation: The evaluation is comprehensive:
Large-scale AI-vs-AI matches (1,000–10,000 games) with statistical significance (z > 5).
Human evaluation across skill levels (100 games vs. players rated 1450–2006, 97% win rate).
A decisive 16–4 victory over the #1 human player, with p = 0.011 (binomial test).

Extensive ablation studies isolating the contribution of each component (e.g., KLUSS alone improves win rate from 58% to 85% vs. ZS21).
Algorithmic Innovation: The combination of one-sided GT-CFR, last-iterate strategy selection, and purification is well-motivated and empirically validated. The use of Stockfish as a node evaluator is pragmatic and effective.

Reproducibility & Transparency: Private game links are provided; hardware requirements are modest (consumer CPU); pseudocode and detailed appendices support reproducibility.

**Weaknesses:**

The article is not clearly articulated and lacks proper formatting. It fails to explicitly highlight the core contributions of the KLUSS algorithm and one-sided GT-CFR (which may represent the most significant departure from prior methods).

Stockfish is designed for chess with perfect information. Using it as an evaluation function for the game tree in FoW chess is not entirely reasonable, since the winning conditions of the two games differ. Please provide an explanation.

**Questions:**

Stockfish is designed for chess with perfect information. Using it as an evaluation function for the game tree in FoW chess is not entirely reasonable, since the winning conditions of the two games differ. Please provide an explanation.

---

> ### Author Response · Authors · 2025-11-22
>
> Thank you for your review.
>
> > The article is not clearly articulated and lacks proper formatting. It fails to explicitly highlight the core contributions of the KLUSS algorithm and one-sided GT-CFR (which may represent the most significant departure from prior methods).
>
> The contributions are also elaborated on in detail in the appendix. See also the general comment to all reviewers above regarding contributions.
>
> > Stockfish is designed for chess with perfect information. Using it as an evaluation function for the game tree in FoW chess is not entirely reasonable, since the winning conditions of the two games differ. Please provide an explanation.
>
> The winning conditions in regular and FoW chess are essentially identical, modulo stalemate being a draw in regular chess and a win in FoW. We agree, though, that regular chess is far from the same game as FoW chess, and that it should be considered somewhat fortunate that the evaluation function from the perfect-information version of the game works well enough for the imperfect-information version. We believe that this is, at least in part, due to the "tactical/short-term" FoW chess: imperfect information is mostly relevant in FoW chess on a tactical (short-term) level and less so on a strategic (long-term) level; as such, combining an imperfect-information search algorithm (which handles the short-term imperfect information concerns) with a long-term evaluation function from perfect-information chess was sufficient for superhuman performance.
>
> The general comment above also contains further comments on the evaluation function and generality of our techniques.

---

### Official Review · Reviewer_qYdi · 2025-11-02

**Soundness:** 3
**Presentation:** 4
**Contribution:** 3
**Rating:** 8
**Confidence:** 4

**Summary:**

This paper presents Obscuro, a search-based AI agent achieving superhuman performance in Fog of War chess, introducing general subgame-solving algorithms that do not rely on explicit common-knowledge reasoning. The authors propose the KLUSS framework that prunes high-order belief states considered strategically irrelevant. This is integrated into algorithms like PCFR+ for efficient tree expansion. Expirical results show that Obscuro surpasses the previous SOTA system and human experts.

**Strengths:**

- The paper addresses a central limitation of existing imperfect-information search methods. KLUSS provides a pragmatic and scalable alternative that extends the reach of search-based techniques.
- The system combines ideas from recent developments in counterfactual regret minimization, tree expansion policies, and real-time planning, demonstrating a coherent and well-engineered design.
- The empirical results are strong. The performance can be achieved with relatively small computation, underscoring the method's practical efficiency.

**Weaknesses:**

- The interaction between PCFR+, one-sided GT-CFR, and KLUSS lacks a unified theoretical analysis. Each component has individual convergence guarantees under specific settings, but their concurrent use in a dynamically expanding and pruned search tree leaves correctness unproven.
- Although framed as a general search method, Obscuro’s performance critically depends on Stockfish’s perfect-information evaluation function. This component embeds extensive domain knowledge from conventional chess, potentially limiting the generality of the claimed framework.
- The paper can also benefit from adding tests on other imperfect-information domains such as poker or Stratego, which strengthens the empirical basis for calling the approach “general-purpose.”

**Questions:**

See weaknesses

---

> ### Author Response · Authors · 2025-11-22
>
> Thank you for your review.
>
> > The interaction between PCFR+, one-sided GT-CFR, and KLUSS lacks a unified theoretical analysis. Each component has individual convergence guarantees under specific settings, but their concurrent use in a dynamically expanding and pruned search tree leaves correctness unproven.
>
>
> KL(U)SS is not safe in theory, so expecting that the combination of PCFR+, one-sided GT-CFR, and KLUSS leads to provable equilibrium play is too much to hope for, despite its strong practical performance. We believe that exploring the theoretical properties of these algorithms, and their combination, is an interesting direction for future work. For example, even the theoretical last-iterate convergence of PCFR+ alone in extensive-form games is already an open problem.
>
>
>
> > Although framed as a general search method, Obscuro’s performance critically depends on Stockfish’s perfect-information evaluation function. This component embeds extensive domain knowledge from conventional chess, potentially limiting the generality of the claimed framework.
>
> Please refer to the general comment above for a discussion of the generalizability of our techniques beyond FoW chess.

---

### Official Review · Reviewer_eLzR · 2025-11-03

**Soundness:** 4
**Presentation:** 4
**Contribution:** 2
**Rating:** 4
**Confidence:** 4

**Summary:**

This paper introduces *Obscuro*, a superhuman Fog of War (FoW) chess agent.
It improves the SOTA by using better SOTA algorithms for the individual steps. The most significant improvement results from replacing LP-based equilibrium computation with GT-CFR. Further improvements are generated by careful engineering.

**Strengths:**

- very good motivation
- great outcomes, producing the first superhuman FoW agent
- overall well-written paper

**Weaknesses:**

- low originality by mainly engineering the current SOTA
- contributions only clearly stated in the ablation
- use of crafted, not learned value-function
- no learning at all, only search
- overuse of footnotes decreases readability

**Questions:**

- Can you motivate the choices of your adaptations?
- What are the implications of your work in the field? I.e., how can your work contribute to solving other tasks?
- The approach uses advanced search techniques and does not utilize any learning algorithm or learnable parameters. What makes this paper a reinforcement learning paper?

---

> ### Comment · Reviewer_abjR · 2025-11-12
> **Search as Focus**
>
> "no learning at all, only search"
> I found this not to be a problem. Search is part of AI.

---

> > ### Comment · Reviewer_eLzR · 2025-11-13
> >
> > While I agree that plain search is part of AI, it is not part of ML, nor is it part of the chosen primary area, RL.
> >
> > The origin of ICLR lies in ML, and my understanding is that it's still focused on learning rather than being a full circle AI conference. Since I'm not entirely certain about this, I'm happy to discuss this and be educated if I'm wrong.
> >
> > However, these are not my main concerns. I just wanted to note it and put it up for discussion.

---

> ### Author Response · Authors · 2025-11-22
>
> Thank you for your review.
>
> We disagree that our work is of "low originality". See the general comment to all reviewers above regarding new techniques that are, to our knowledge, only found in this paper. In the end, our results are a significant improvement over the prior SOTA, which was below the level of top humans while ours is significantly above.
>
> > overuse of footnotes decreases readability
>
> We have moved some of the footnotes into the main text to help readability.
>
> > Can you motivate the choices of your adaptations?
>
> In what is now Appendix C, which goes into detail on each of the improvements that we have made, we give motivation for each improvement. If the reviewer has any specific question, we would be happy to answer or clarify.
>
> > What are the implications of your work in the field? I.e., how can your work contribute to solving other tasks?
>
> Please refer to the general comment above for a discussion of the generalizability of our techniques beyond FoW chess.
>
> > The approach uses advanced search techniques and does not utilize any learning algorithm or learnable parameters. What makes this paper a reinforcement learning paper?
>
> Multiagent RL (namely, no-regret learning via PCFR+) is used to solve subgames, so it is not the case that no learning is involved in the paper. There are many papers every year in ICLR that have no-regret learning in games as the primary connection to reinforcement learning; ours is no exception.

---

### Official Review · Reviewer_74mV · 2025-11-04

**Soundness:** 2
**Presentation:** 1
**Contribution:** 2
**Rating:** 2
**Confidence:** 4

**Summary:**

This paper contributes a Fog-of-War chess bot that is ostensibly state of the art. It performs real-time search without trying to untangle unwieldly "I know that you know..." loops that usually choke imperfect-information solvers.

**Strengths:**

Obscuro seems to be the first superhuman AI in the fog of war chess variant.
The paper does a good job of explaining the difficulties associated with developing AI in this kind of setting.

**Weaknesses:**

- My biggest concern is the significance of the contribution. Even if all of the claims in the paper are completely accurate (and see below for concerns on that front), it's not clear to me that applying mostly known tricks to develop superhuman AI in this niche chess variant constitutes enough for acceptance. Despite being a lifelong chess fan, and even a chess variants fan, I've never heard of fog of war chess. I don't think the abstract's claim that FoW chess has been "the main challenge problem in imperfect information games" is accurate.

- Building on the previous point, the methods used are interesting but not super exciting, and don't seem to be very generalizable. The work would be stronger if the authors could demonstrate or at least argue why their innovations are useful beyond FoW chess.

- The results section is quite short, and I'm not convinced that the authors have done everything they can to put Obscuro up to the toughest tests.

- The paper is somewhat unprofessionally written (e.g. section 4.1; the 4-line title).

**Questions:**

- What is the broader significance?
- How could the methods generalize?

---

> ### Comment · Reviewer_abjR · 2025-11-12
> **Stratego?**
>
> Would you accept the paper if the authors added another variant like Stratego?

---

> ### Author Response · Authors · 2025-11-22
>
> Thank you for your review.
>
> Please see the general comment to all reviewers for our comments regarding originality and generalizability beyond FoW chess.
>
>
> > Despite being a lifelong chess fan, and even a chess variants fan, I've never heard of fog of war chess.
>
> This is a bit of a surprise to us---on chess.com, at least, it's one of the most popular variants. As of the writing of this text, it is [hovering around Crazyhouse and Giveaway (Losing) chess](https://www.chess.com/variants) in terms of active logged-in players. Earlier this year, chess.com also hosted a [tournament with prize money](https://www.chess.com/news/view/2024-chesscom-fog-of-war-chess-championship-knockout-chan-wins). Furthermore, it is currently by far the most popular form of imperfect-information chess.
>
> > I don't think "the main challenge problem in imperfect information games" is accurate.
>
> We have weakened this to "a major challenge problem". For example, we are aware of other research groups who were trying to achieve superhuman performance in FoW chess, so it was a recognized ambitious milestone.
>
> We disagree that our work is "applying mostly known tricks". As other reviewers pointed out, both KLUSS and one-sided GT-CFR, at minimum, are notable departures from previous work that demonstrate practical gains with statistical significance.
>
>
> > The results section is quite short, and I'm not convinced that the authors have done everything they can to put Obscuro up to the toughest tests.
>
> A 20-game sample, although seemingly small, was enough for us to establish a convincing 16-4 victory over the top human under conditions most familiar to the top human (most standard time control, using the standard chess.com interface, etc.). The games were also played over the course of two days, 10 games per day---so our human opponent can (and did) analyze the first set of games games overnight. We have included a note about this in the revised version.
>
> > The paper is somewhat unprofessionally written (e.g. section 4.1; the 4-line title).
>
>
> We  have revised some parts of Section 4.1 to fix some minor errors (see responses to other reviewers) and improve readability. We have uploaded a corresponding new version of the paper. We welcome any further suggestion regarding writing improvements to this section or to the title.

---

### Author Response · Authors · 2025-11-22
**General comment to all reviewers**

**Contributions and new techniques in our paper**

Some reviewers found our contributions to be "low originality", "mostly known", or not novel. We disagree. As some reviewers have pointed out, our paper includes several techniques that lead to practical improvements and are, to our knowledge, novel---including new variants of knowledge-limited subgame solving and GT-CFR+, and more. For space reasons, it is impossible to include all the detail in the main body, as it would require a large amount of notation and setup (see, for example, the exposition of extensive-form games that is set up in the appendix, which is two pages long and is required to go into detail about the techniques). We opted instead to use the main body to set up the problem of interest, discuss the difficulties that FoW chess introduces over prior superhuman game-playing breakthroughs, and to give a higher-level overview of our techniques. We have added wording to Section 3 to make this organizational choice clearer.

**Generalizability of our techniques beyond FoW chess**

Multiple reviewers left comments asking, in various ways, about how our techniques may generalize beyond FoW chess. Besides the use of a chess-specific state value function (from Stockfish), all our techniques are generalizable to any two-player zero-sum imperfect-information game. For example, KLUSS and one-sided GT-CFR are drop-in replacements for their respective predecessors, that we would expect to lead to improved performance wherever their predecessors may be applied, i.e., in any two-player zero-sum imperfect-information game.

Regarding the chess-specific evaluation function, it is of course true that such a function is not always available. For a more general game, one could substitute this evaluation function for one that is learned from data or self-play. However, because our main focus in the present paper is on search, and because Stockfish's evaluation was sufficient for superhuman performance, we did not feel the need to pursue this direction.

---

### Meta-Review · Area_Chair_kmD9 · 2026-01-08

**Summary:**

Reviewers praised the achievement of superhuman performance in Fog of War chess and the empirical strength of the proposed search techniques, but raised issues with originality, generalizability beyond chess, reliance on Stockfish's evaluation function, lack of theoretical analysis for combined methods, and presentation clarity. These concerns shaped a recommendation for acceptance as a poster, balancing the solid results against calls for broader applicability and refinements.

**Reviewer Concerns:**

The rebuttal addressed originality by highlighting novel elements like KLUSS and one-sided GT-CFR, and clarified generalizability to other imperfect-info games. It also explained Stockfish's use as pragmatic for search focus, with potential for learned alternatives. Outstanding issues include limited testing on non-chess domains like poker or Stratego, absence of unified theoretical guarantees, and some unresolved clarity problems in figures and ablations.

**Reviewer Scores:**

Reviewer 74mV might raise from 2 to 4, acknowledging rebuttal on novelty but still doubting broader impact. Reviewer eLzR could bump to 6, as generalizability and motivations were clarified. Reviewer qYdi stays at 8. Reviewer 3cmb might increase to 7 with presentation fixes noted. Reviewer abjR remains at 8. Reviewer uPu3 holds at 8, with some clarity addressed.

---

### Decision · Program_Chairs · 2026-01-26

Accept (Poster)